# *N*-acetylneuraminic acid links immune exhaustion and accelerated memory deficit in diet-induced obese Alzheimer's disease mouse model

Stefano Suzzi [1,6] ✉, Tommaso Croese [1,6], Adi Ravid[2], Or Gold[2], Abbe R. Clark[3,4], Sedi Medina[1], Daniel Kitsberg[2], Miriam Adam[2], Katherine A. Vernon[3,4], Eva Kohnert[3,4], Inbar Shapira [2], Sergey Malitsky [5], Maxim Itkin [5], Alexander Brandis[5], Tevie Mehlman[5], Tomer M. Salame[5], Sarah P. Colaiuta [1], Liora Cahalon[1], Michal Slyper[3,4], Anna Greka [3,4,7] ✉, Naomi Habib [2,7] ✉ & Michal Schwartz [1,7] ✉

Systemic immunity supports lifelong brain function. Obesity posits a chronic burden on systemic immunity. Independently, obesity was shown as a risk factor for Alzheimer's disease (AD). Here we show that high-fat obesogenic diet accelerated recognition-memory impairment in an AD mouse model (5xFAD). In obese 5xFAD mice, hippocampal cells displayed only minor diet-related transcriptional changes, whereas the splenic immune landscape exhibited aging-like CD4[+] T-cell deregulation. Following plasma metabolite profiling, we identified free *N*-acetylneuraminic acid (NANA), the predominant sialic acid, as the metabolite linking recognition-memory impairment to increased splenic immune-suppressive cells in mice. Single-nucleus RNA-sequencing revealed mouse visceral adipose macrophages as a potential source of NANA. In vitro, NANA reduced CD4[+] T-cell proliferation, tested in both mouse and human. In vivo, NANA administration to standard diet-fed mice recapitulated high-fat diet effects on CD4[+] T cells and accelerated recognition-memory impairment in 5xFAD mice. We suggest that obesity accelerates disease manifestation in a mouse model of AD via systemic immune exhaustion.

Alzheimer's disease (AD) is the most common form of dementia, characterized by progressive brain amyloidosis and neuronal loss. For decades regarded as neuron-centric, AD pathogenesis is now recognized as being strongly affected by the state of the whole organism[1,2]. In particular, a healthy immune system is required to support brain maintenance and repair[3–11], whereas immune aging increases AD risk[12–15]. Among the many age-related changes in the immune system is immune exhaustion, marked by the progressive loss of immune effector functions, decrease in cytokine production, and increased inhibitory immune checkpoint signaling[16–18]. As such, exhausted

[1]Weizmann Institute of Science, Department of Brain Sciences, Rehovot, Israel. [2]The Hebrew University of Jerusalem, Edmond & Lily Safra Center for Brain Sciences, Jerusalem, Israel. [3]Broad Institute of MIT and Harvard, Cambridge, MA, USA. [4]Department of Medicine, Brigham and Women's Hospital and Harvard Medical School, Boston, MA, USA. [5]Weizmann Institute of Science, Life Sciences Core Facilities, Rehovot, Israel. [6]These authors contributed equally: Stefano Suzzi, Tommaso Croese. [7]These authors jointly supervised this work: Anna Greka, Naomi Habib, Michal Schwartz. ✉e-mail: stefano.suzzi@weizmann.ac.il; agreka@broadinstitute.org; naomi.habib@mail.huji.ac.il; michal.schwartz@weizmann.ac.il

immune cells have a reduced ability to protect the brain. In line with this contention, immune deficiency as well as immune suppression and exhaustion were linked to disease severity in mouse models of AD, while reducing or blocking immune suppression or exhaustion was found to attenuate pathological manifestations[19–25]. Based on this cumulative evidence, we envisioned that environmental conditions that are known as strong risk factors for AD might promote disease manifestations by negatively affecting immune functions. One such environmental condition is obesity, which is among the strongest AD risk factors and the most frequent AD comorbidity[26–31]. Of note, obesity is associated with persistent systemic inflammation and impaired immune responses that have been linked to increased risk of infections and other chronic conditions such as allergy and cancer[32].

Here, we specifically focused on diet-induced obesity, and hypothesized that it might affect AD by deregulating systemic immunity. Using a transgenic mouse model carrying five human AD-linked mutations (5xFAD), we found that obesity induced by high-fat diet accelerated the onset of disease manifestations, including recognition-memory impairment, which was associated with increased splenic levels of exhausted CD4$^+$ T effector memory cells, CD4$^+$FOXP3$^+$ regulatory T cells, and increased blood levels of the metabolite $N$-acetylneuraminic acid. In vitro and in vivo studies revealed that this metabolite could induce immune exhaustion and accelerate recognition-memory impairment in 5xFAD mice.

## Results

### Diet-induced obesity accelerated disease manifestations in 5xFAD mice

To test the impact of obesity on AD pathology, we generated a mouse model of obesity-AD comorbidity using 5xFAD mice, a transgenic model of amyloidosis[33], that were fed with high-fat obesogenic diet (HFD) or standard control diet (CD). Wild-type (WT) mice fed with either HFD or CD were also included in the study. Both female and male mice were used. Since mid-life obesity increases the risk of AD[29–31], we chose a long-term diet regimen starting from around 2 up to 8-9 months of age (mo) for a total of 24-28 weeks (Fig. 1a). Since the metabolic responses to HFD were similar between females and males (Supplementary Fig. 1a–h), we combined both sexes for subsequent analyses. We used the novel object recognition (NOR) test to assay recognition-memory performance (Fig. 1b), known to decline in the 5xFAD model[34,35]. The NOR test, which allows repeated measures, was used for a longitudinal follow-up to determine if and when the HFD accelerated cognitive impairment in 5xFAD mice. At the age of 6.5 mo, both CD and HFD-fed 5xFAD mice maintained intact performance in the NOR test (Supplementary Fig. 2a, b). At 8 mo, HFD-fed 5xFAD mice showed loss of NOR capability, whereas CD-fed 5xFAD mice still performed similarly to WT controls fed with CD or HFD (Fig. 1c; Supplementary Fig. 2c). HFD did not affect the performance in the NOR test of WT mice at either time point (Supplementary Fig. 2a; Fig. 1c). Locomotor activity was not different between groups (Supplementary Fig. 2d, e), whereas the time spent in the middle of the arena was lowest for HFD-fed WT mice, possibly indicating slightly higher anxiety levels in HFD-fed WT relative to CD-fed WT mice, but not in HFD-fed 5xFAD relative to CD-fed 5xFAD mice (Supplementary Fig. 2f, g).

After the last repeated NOR test, in which we detected that the HFD-fed 5xFAD mice had lower score than the CD-fed 5xFAD mice, we euthanized the mice and assessed if and how the obesogenic diet affected brain pathology. Specifically, we tested typical AD-related hallmarks in the 5xFAD model[20–22] (Fig. 1d). No changes in neuronal survival or reactive astrogliosis were noticeable in HFD-fed WT mice compared to CD-fed controls (Supplementary Fig. 3a–c). In 5xFAD mice, the HFD did not affect the level of amyloid β oligomers (Aβ1-42; Supplementary Fig. 3d–f) nor amyloid β plaques (Fig. 1e, f; Supplementary Fig. 3g, h). However, compared to CD-fed 5xFAD mice, the HFD-fed 5xFAD mice displayed increased neuronal loss (subiculum,

statistically significant, Fig. 1g, h; cortex layer V, trend, Supplementary Fig. 3i, j) and earlier manifestation of astrogliosis (Fig. 1i, j). Altogether, these results indicate that the HFD accelerated the onset of disease manifestations in 5xFAD mice but did not affect any of the other tested parameters in otherwise healthy age-matched WT mice.

### AD and HFD had largely independent effects on the mouse hippocampal cell fate

To closely analyze the brain's cellular landscape, we performed single-nucleus RNA-sequencing of the hippocampus (sNuc-Seq, 10x genomics, Methods; Fig. 2a; Supplementary Fig. 4a–f, 5a–g). Briefly, we found effects of both morbidities, most prominently AD-associated alterations in microglia and astrocytes in 5xFAD mice, as previously reported[36,37], and an HFD-associated increase in oligodendrocytes that was more prominent in WT mice (Supplementary Fig. 4f). In addition, we found a cluster of cells within the dentate gyrus (DG1) that was specifically over-represented in HFD-fed 5xFAD mice (Fig. 2b, c; Supplementary Fig. 5a). Gene set enrichment analysis of the differentially expressed genes in this cluster highlighted several pathways related to neuronal differentiation, integration, and growth (hypergeometric test, FDR < 0.050, Fig. 2d; Supplementary Data 1), indicative of an immature-like neuronal phenotype.

Profiling of the major non-neuronal cell populations (microglia and other immune cell types, astrocytes, and oligodendrocytes) identified a range of glial cell states, including distinct states associated with AD and HFD (Fig. 2e–j). Sub-clustering analysis of microglia revealed the expected AD-associated decrease in homeostatic microglia (HMG, $Cx3cr1^{high}P2ry12^{high}$ cells) and the elevation of disease-associated microglia[36] (DAM, $Trem2^{high}Cd9^{high}$ cells), but no further HFD-related effects on microglial sub-states and frequencies (Fig. 2e, f). Within the non-microglia immune cell compartment, we detected a small population of perivascular macrophages (PVMs, $Mrc1^+F13a1^+$ cells) that significantly decreased with AD (Fig. 2e, f). In addition, we found a small population of T cells ($Trbc2^+Cd4^+$ cells; differentially upregulated genes compared to all other immune cell clusters in Supplementary Data 2) that was barely detectable in WT mice, but was found in 5xFAD mice and almost doubled in HFD-fed 5xFAD relative to CD-fed 5xFAD mice (Fig. 2e, f). In the astrocyte compartment, sub-clustering analysis revealed prominent AD-associated effects. In line with a previous report[38], we identified two $Gfap^{low}$ homeostatic states (AST1 and AST2) and two $Gfap^{high}$ states, including the previously described disease-associated astrocytes[37] (DAAs, $Serpina3n^+$ cells; Fig. 2g, h). AST1 cells ($Mfge8^{high}Garem1^{high}$) were markedly reduced in 5xFAD mice of both diet groups, concomitantly with the increase in DAAs (Fig. 2g, h). In the oligodendrocyte compartment, we identified AD-associated alterations of distinct cell subsets, such as the decrease in $Ptgds^{high}$ mature oligodendrocytes[39,40] (OLG1) and the prominent increase in disease-associated oligodendrocytes[41,42] (DOLs, $C4b^+$ cells) and committed oligodendrocyte precursors (COPs; Fig. 2i, j).

We next tested the differentially expressed genes (DEGs) across all cell types between HFD-fed and CD-fed 5xFAD mice (Fig. 2k–m). The strongest response was found in astrocytes (Fig. 2l); microglia and oligodendrocytes also showed a differential transcriptional response to HFD, albeit comparably lower DEGs (Fig. 2k, m). Notably, both microglia and astrocytes of HFD-fed 5xFAD mice displayed reduced expression of $Apoe$, encoding apolipoprotein E (APOE; Fig. 2k, l, bold). In humans, APOE is involved in the clearance of amyloid β and other protein aggregates, and its variant APOE4, which is less efficient in clearing amyloid β, is the major genetic risk factor for AD[43,44]. While the observed transcriptional response of neuronal and glial cells might contribute to comorbidity, the overall HFD effect on the cellular landscape of the hippocampus did not seem to be robust to the extent that it could explain the accelerated disease progression in HFD-fed 5xFAD mice, which prompted us to search for a possible HFD-induced effect outside the brain.

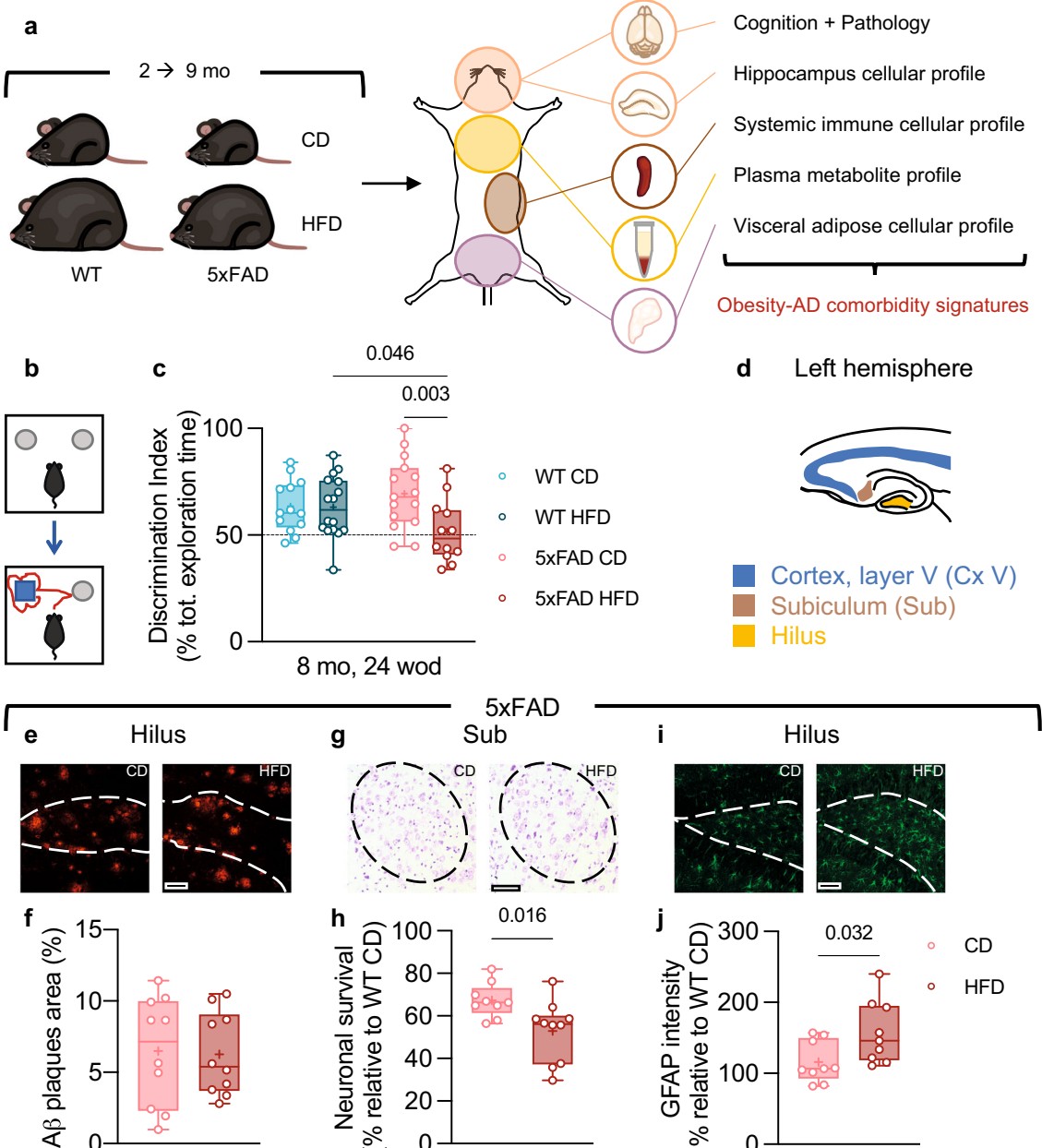

**Fig. 1 | High-fat diet accelerated disease manifestations relative to control diet in 5xFAD mice. a** Overview of the experimental strategy. **a, c** Reduction in novelty discrimination in HFD-fed 5xFAD mice. Evaluation of cognition with the NOR test (**b**) at 8 mo, 24 weeks of diet (wod; **c**). Mice from two independent experiments, sample $n$: WT CD = 13, WT HFD = 16, 5xFAD CD = 15, 5xFAD HFD = 12. Statistical analyses: two-way ANOVA followed by Fisher's LSD post hoc test. **d**, Brain regions analyzed for histopathology. **e, f** Assessment of Aβ plaques load in the hippocampal hilus of 5xFAD mice. **g, h** Neuronal survival in the subiculum (Sub) of 5xFAD mice. **i, j**, Glial fibrillary acidic protein (GFAP)-immunoreactivity in the hippocampal hilus of 5xFAD mice. **e, g, i** Representative images, left: CD, right: HFD, scale bars: 70 μm. **f, h, j** Mice from two independent experiments, sample $n$: **f**, 5xFAD CD = 10, 5xFAD HFD = 10; **h**, 5xFAD CD = 9, 5xFAD HFD = 10; **j**, 5xFAD CD = 9, 5xFAD HFD = 9. **h** Data normalized by average WT CD value (Supplementary Fig. 3b). **j** Data normalized by average WT CD value (Supplementary Fig. 3c). Statistical analyses: two-tailed unpaired Student's $t$ test. **c, f, h, j** Box plots represent the minimum and maximum values (whiskers), the first and third quartiles (box boundaries), the median (box internal line), and the mean (cross). Statistical analyses: two-way ANOVA followed by Fisher's LSD post hoc test. Source data are provided as a Source Data file.

## HFD induced immune exhaustion in 5xFAD mice

To test our working hypothesis that HFD-induced obesity might accelerate disease manifestations via imposing systemic immune deregulation, we first analyzed the circulating immune cell landscape by mass cytometry (CyTOF), and found that the major effect was a trend of an increase of CD4⁺ T cells in the HFD-fed group in both WT and 5xFAD mice (Supplementary Fig. 6a–c). In subsequent experiments, we focused on the lymphocyte compartment in the spleen. Of note, deregulation within the splenic CD4⁺ T-cell compartment in mice was previously linked to aging[45–47] and neurodegeneration[20,48]. Thus,

we analyzed freshly isolated splenocytes from mice culled after 28 weeks of diet by flow cytometry (Supplementary Fig. 7a). While CD4⁻ T cells were unaffected (Supplementary Fig. 7b–d), we found shrinkage of the naïve CD4⁺ T-cell population (Fig. 3a) and expansion of CD4⁺ T effector memory cells (TEMs; Fig. 3b) and CD4⁺FOXP3⁺ regulatory T cells (Tregs; Fig. 3c), all features associated with immune aging in mice[45–47]. Based on these findings, we used mass cytometry (CyTOF) to investigate more closely the profile of the CD4⁺ T-cell compartment, using mouse splenocytes that were cryopreserved at the end of the NOR follow-up (Supplementary Fig. 8a–c).

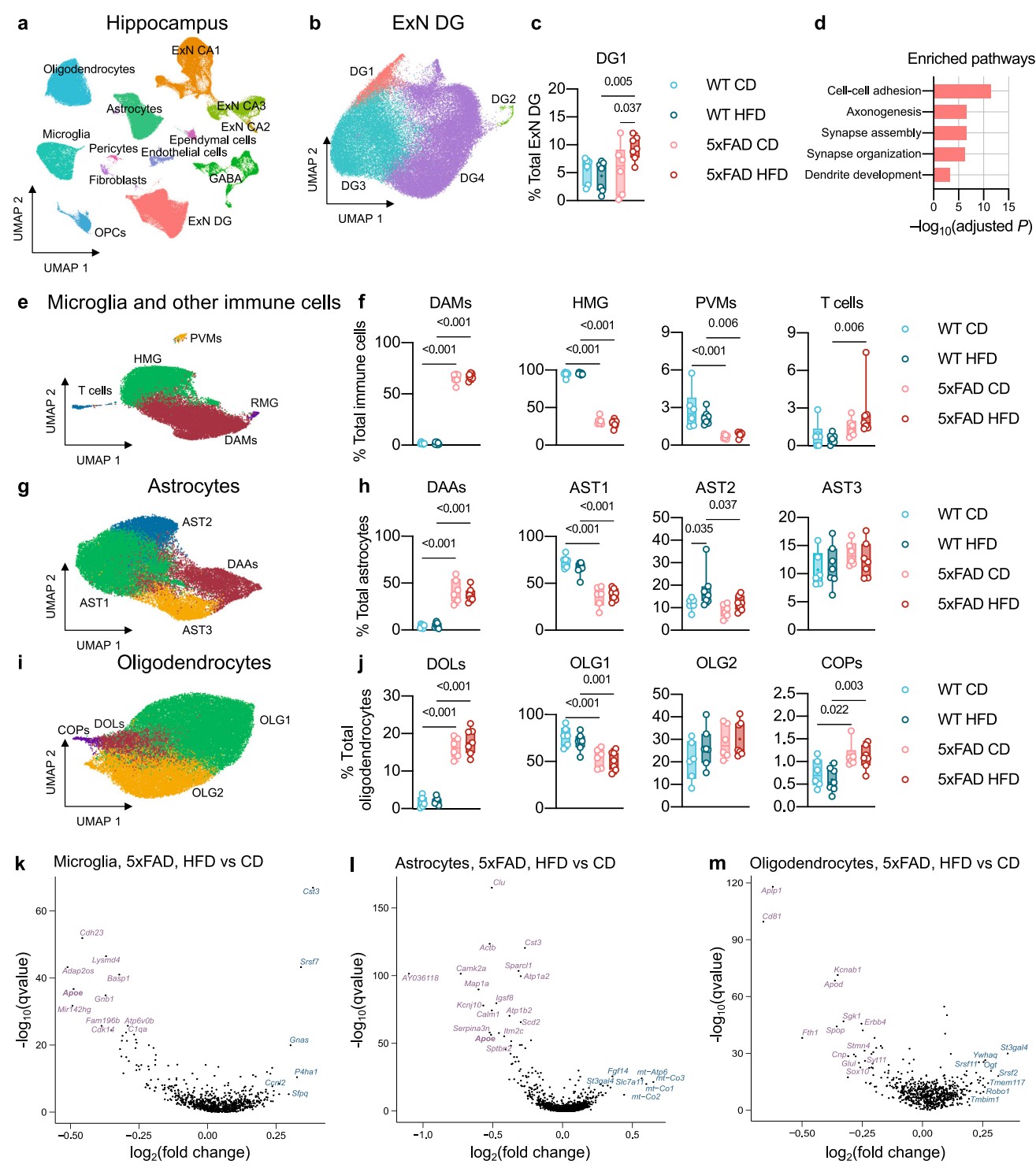

Sub-clustering analysis of the CD4+CD44high TEM population identified 6 distinct clusters (Fig. 3d–f; Supplementary Fig. 9a). HFD-fed 5xFAD mice displayed the highest frequencies of exhausted TEMs (Cluster 6), characterized by increased expression of exhaustion markers, including the inhibitory immune checkpoint receptors PD-1, TIGIT and LAG-3 and the transcription factors TOX and EOMES, and reduced expression of tested cytokines[16–18] (Fig. 3e, f). Furthermore, analysis of the expression of the same exhaustion markers over the entire CD4+ TEM compartment revealed their specific increase in HFD-fed 5xFAD mice (Supplementary Fig. 9b). Overall, the systemic immune system of 5xFAD mice showed augmented susceptibility to the HFD-induced challenges relative to age-matched WT mice, which resulted in

increased splenic frequency of immune-suppressive Tregs and exhausted TEMs.

## Plasma N-acetylneuraminic acid linked recognition-memory impairment and elevation of splenic Tregs in mice

We hypothesized that diet-induced metabolites might be responsible for the specific effect of HFD on the immune system of 5xFAD mice. To address this question, we carried out plasma metabolite profiling collected from the mice that were evaluated for both NOR and systemic immune phenotype. We identified a total of 229 metabolites (Supplementary Data 3). For each of these metabolites, a Cell Means Model was used to single out the metabolites that most closely

**Fig. 2 | AD and HFD-related changes in the cellular landscape of the mouse hippocampus. a** UMAP embedding of 237,631 single nuclei profiles (sNuc-Seq), colored after post hoc cell type annotation. Mice from five independent experiments, sample $n = 28$. For quantifications and statistical analyses, 219,237 nuclei were included from $n = 26$ samples: WT CD = 6, WT HFD = 7, 5xFAD CD = 6, 5xFAD HFD = 7 (Methods, *Cell fraction estimations and statistics* section). CA1–3, *cornu Ammonis* region 1–3; DG, dentate gyrus; ExN, excitatory neurons; GABA, GABAergic neurons; OPCs, oligodendrocyte precursor cells. **b, c** Sub-clustering analysis of the DG granule neurons (ExN DG). Sample $n$: see **a. b** UMAP embedding of sNuc-Seq profiles colored by cluster. **c** Changes in frequency of DG1 cluster across experimental conditions; DG2–4 clusters are shown in Supplementary Fig. 5a. **d** Pathway analysis of the genes associated with DG1 showing enrichment of pathways related to neuronal differentiation, integration, and growth (FDR-adjusted hypergeometric test $P$-value <0.050). **e–j** Sub-clustering analysis of the brain's immune cells including microglia (**e, f**), astrocytes (**g, h**), and oligodendrocytes (**i, j**). Sample $n$: see **a. e, g, i** UMAP embedding of sNuc-Seq profiles colored by cluster.

**f, h, j** Changes in frequency of cell types across experimental conditions. Abbreviations: AST1–3, astrocyte clusters 1–3; COPs, committed oligodendrocyte precursors; DAAs, disease-associated astrocytes; DAMs, disease-associated microglia; DOLs, disease-associated oligodendrocytes; HMG, homeostatic microglia; OLG1, 2, oligodendrocyte clusters 1, 2; PVMs, perivascular macrophages; RMG, replicating microglia. **c, f, h, j** Box plots represent the minimum and maximum values (whiskers), the first and third quartiles (box boundaries), the median (box internal line), and the mean (cross). Statistical analyses: two-way ANOVA followed by Fisher's LSD post hoc test. Source data are provided as a Source Data file. **k–m** HFD-induced gene expression programs in microglia (**k**), astrocytes (**l**), and oligodendrocytes (**m**) in 5xFAD mice. Volcano plot representation of differentially expressed genes in HFD-fed 5xFAD mice ($n = 7$) compared to CD-fed 5xFAD controls ($n = 6$). *Apoe*, downregulated in both microglia and astrocytes, is highlighted in bold. X-axis: average $\log_2$ fold change (HFD relative to CD); Y-axis: FDR-adjusted MAST test $P$-value <0.010 ($-\log_{10}$).

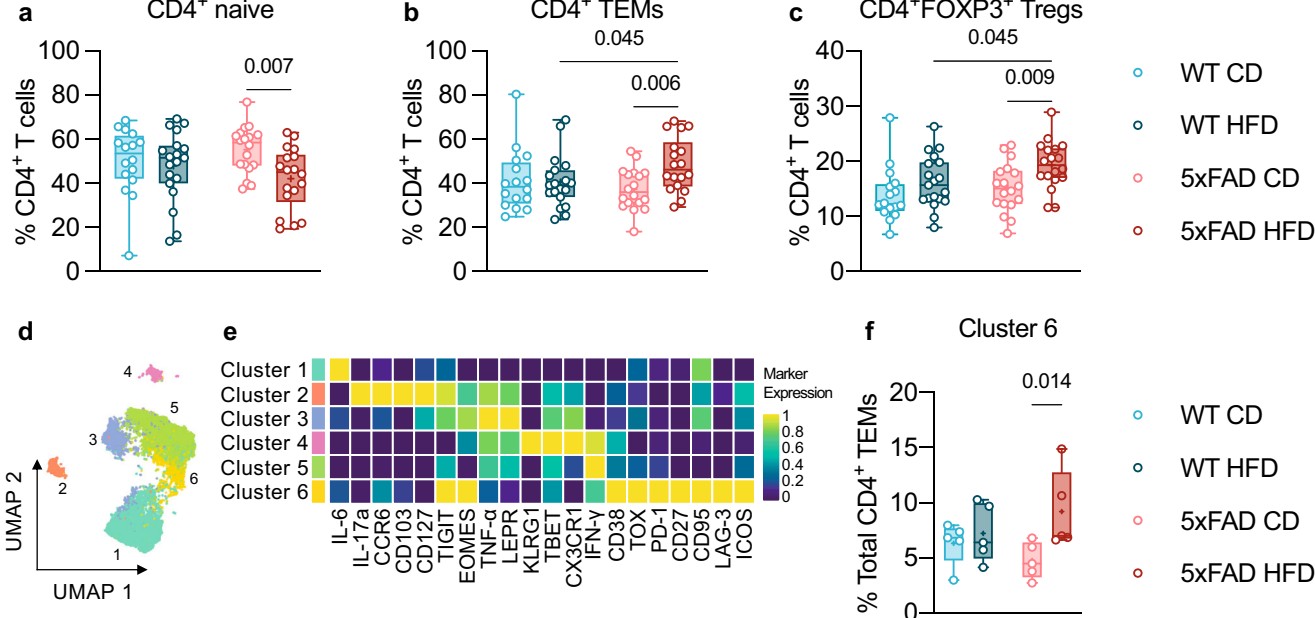

**Fig. 3 | Splenic CD4[+] T-cell rearrangements in HFD-fed 5xFAD mice. a–c** CD4[+] T-cell immune deviations in HFD-fed 5xFAD mice. Flow cytometric quantification of splenic frequencies of CD4[+] naive T cells (CD44[low]CD62L[high]; **a**), CD4[+] TEMs (CD44[high]CD62L[low]; **b**), and CD4[+]FOXP3[+] Tregs (**c**). Mice from two independent experiments, sample $n$: WT CD = 16, WT HFD = 19, 5xFAD CD = 18, 5xFAD HFD = 18. Statistical analyses: two-way ANOVA followed by Fisher's LSD post hoc test. **d–f** Characterization of the CD4[+] TEM compartment by mass cytometry. Cryo-preserved splenocytes from mice evaluated for both cognition (NOR test; Fig. 1c) and systemic immune phenotype (Fig. 3a–c) were used for CyTOF analysis of the CD4[+] TEM compartment, sample $n$: WT CD = 5, WT HFD = 5, 5xFAD CD = 5, 5xFAD

HFD = 5. **d** UMAP embedding of CD4[+] TEM cell clusters (2000 cells, randomly selected from each animal). FlowSOM-based immune cell populations are overlaid as a color dimension. **e** Mean population expression levels of markers used for UMAP visualization and FlowSOM clustering of CD4[+] TEMs. **f** Increased frequency of exhausted TEMs in HFD-fed 5xFAD mice. Sub-clustering analysis of the CD4[+] TEM compartment identified six clusters; Cluster 6 only is shown here, Clusters 1 to 5 are shown in Supplementary Fig. 9a. Statistical analyses: two-way ANOVA followed by Fisher's LSD post hoc test. **a–c, f** Box plots represent the minimum and maximum values (whiskers), the first and third quartiles (box boundaries), the median (box internal line), and the mean (cross). Source data are provided as a Source Data file.

associated with each experimental condition (WT:CD, WT:HFD, 5xFAD:CD, and 5xFAD:HFD; Methods). We identified 46 metabolites with significant differential levels between conditions (one-way *omnibus* ANOVA test $P$-value <0.050; Fig. 4a). Of these 46 metabolites, 22 were significant ($P$-value <0.050; Fig. 4a, asterisks) for the 5xFAD:HFD condition, and were further tested for potential association with the NOR discrimination index and the levels of Tregs in the spleen (Fig. 4b), a parameter previously found to associate with disease severity in mouse models of AD[20,48]. Of these 22 metabolites, free *N*-acetylneuraminic acid (NANA; Supplementary Fig. 10a, b), the predominant form of sialic acid in mammals[49], was not only specifically elevated in the blood of HFD-fed 5xFAD mice (among other metabolites, Fig. 4a; Supplementary Fig. 10c), but it also showed the highest correlation with worsening NOR capability and with

increased splenic Tregs frequency (Fig. 4b–d). Using liquid chromatography with tandem mass spectrometry to assess potential changes of free NANA levels in the hippocampus, we found no differences across groups (Supplementary Fig. 10d, e). Validation on a larger number of plasma samples, which included samples from the same animals used for metabolite profiling, confirmed that circulating free NANA was specifically elevated in HFD-fed 5xFAD mice (Fig. 4e). Using this larger cohort of samples, we found that the correlation between NANA levels and NOR discrimination index (Supplementary Fig. 10f) and between NANA levels and splenic Tregs frequency (Supplementary Fig. 10g) confirmed our findings from metabolite profiling. Based on these results, we further focused on NANA as the metabolite potentially driving the accelerated disease manifestations in HFD-fed 5xFAD mice.

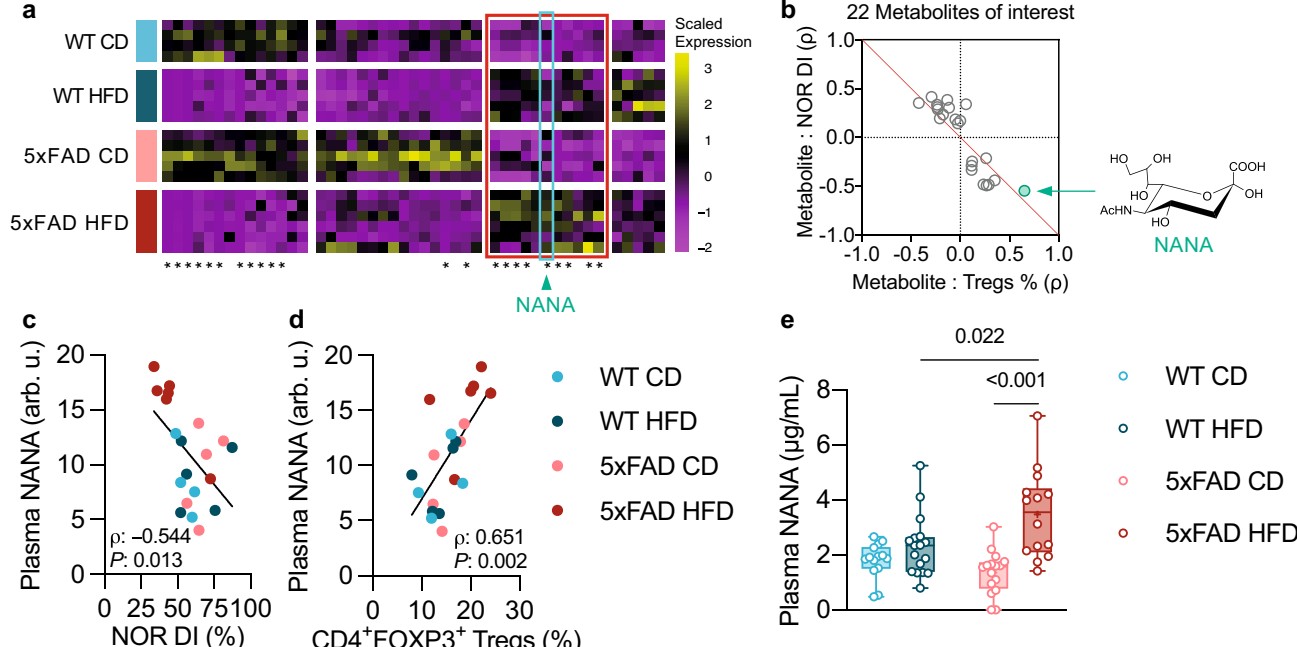

**Fig. 4 | High plasma levels of free NANA in HFD-fed 5xFAD mice. a–d** Metabolite profiling of plasma samples collected from some of the male mice that were evaluated for both cognition (NOR test; Fig. 1c) and systemic immune phenotype (Fig. 3a–c), sample *n*: WT CD = 4, WT HFD = 5, 5xFAD CD = 5, 5xFAD HFD = 6. **a** Heatmap representation of the plasma metabolites whose linear regression models had unadjusted one-way *omnibus* ANOVA test *P*-value <0.050 (46 out of 229 total identified metabolites). Each column represents one metabolite and each row one sample (mouse). Asterisks indicate the metabolites of interest (22 in total; Methods). The red box highlights the block of metabolites whose overall levels trended highest in HFD-fed 5xFAD mice, which include *N*-acetylneuraminic acid (NANA; cyan box and green arrowhead). Complete list of identified metabolites, regression coefficients, and exact *P*-values is provided in Supplementary Data 3, "cell means model" tab. **b**, ρ-ρ plot (Methods). The red diagonal discriminates between metabolites associated with high NOR discrimination index (DI) and low

splenic Tregs abundance (% out of total CD4[+] T cells; quadrant II) and metabolites associated with low NOR DI and high splenic Tregs % (quadrant IV). The position of NANA (inset) is indicated by the green arrow. **c**, **d** Simple linear regression (black line) and Spearman's rank correlation (ρ coefficient, two-tailed *P*-value) between NANA levels, as quantified after plasma metabolite profiling, and NOR discrimination index (DI; **c**) and splenic Tregs abundance (**d**); arb. u., arbitrary unit (normalized peak area/100,000). **e** Quantification of plasma NANA using a fluorometric assay of both female and male mice that were evaluated for both cognition (NOR test; Fig. 1c) and systemic immune phenotype (Fig. 3a–c), also including the same animals described in (**a**–**d**), sample *n*: WT CD = 14, WT HFD = 17, 5xFAD CD = 16, 5xFAD HFD = 14. Statistical analyses: two-way ANOVA followed by Fisher's LSD post hoc test. Box plots represent the minimum and maximum values (whiskers), the first and third quartiles (box boundaries), the median (box internal line), and the mean (cross). **b**–**e** Source data are provided as a Source Data file.

## Visceral fat is a potential source of NANA in HFD-fed 5xFAD mice

To identify a putative source of NANA, we studied the cellular landscape of the most likely target organ of the HFD in the periphery: the visceral adipose tissue (VAT). To this end, we analyzed the fate of the gonadal VAT by sNuc-Seq (10x genomics, Methods; Supplementary Fig. 11a–e, 12a–i). The most prominent effect among the four experimental groups was the obesity-driven expansion of macrophages, similarly in both genotype groups (Supplementary Fig. 11a–e; Fig. 5a, b). In particular, sub-clustering analysis revealed a macrophage population (MAC3) that selectively expanded with HFD, characterized by the simultaneous expression of *Trem2* and several additional markers consistent with the previously described lipid-associated macrophages[50] (LAMs; Supplementary Fig. 12g, h). In contrast, AD alone hardly showed discernible effects.

Given the HFD-induced expansion of macrophages in the VAT, we evaluated the expression profiles of the four neuraminidase *Neu* genes[51], encoding enzymes known to generate NANA. We found that *Neu1* was most abundantly expressed by immune cells, and preferentially by macrophages; in contrast, *Neu3* was hardly detectable, whereas *Neu2* and *Neu4* were below detection level (Fig. 5c). In obese mice, the majority of *Neu1*+ macrophages were MAC3/LAMs (Supplementary Fig. 12i), suggesting that neuraminidase activity might be an important component in the macrophage response associated with disrupted lipid homeostasis[50]. Remarkably, *Neu1*-expressing macrophages were particularly increased in HFD-fed 5xFAD mice (average 3.1-fold relative to CD-fed 5xFAD mice) compared to HFD-fed WT mice

(average 1.7-fold relative to CD-fed WT mice; Fig. 5d). Taken together, these findings suggest that the increased proportion of *Neu1*-expressing macrophages in the VAT combined with the increase of adiposity might account, at least in part, for the specific elevation of circulating NANA levels in HFD-fed 5xFAD mice.

## NANA drove T-cell deregulation in vitro and in vivo and accelerated recognition-memory impairment in 5xFAD mice

Since the major HFD-related changes in 5xFAD mice were in the CD4[+] T-cell compartment, we tested if NANA alone could recapitulate this effect, and if so whether it would also lead to an effect on NOR performance. To test the effect of NANA on T cells, we evaluated T-cell proliferation in both mouse splenic T cells (Supplementary Fig. 13a) and peripheral-blood human T cells (Fig. 6a), cultured in vitro in the presence or absence of NANA. In both mice (Supplementary Fig. 13b–g) and humans (Fig. 6b, c; Supplementary Fig. 14a–e), NANA suppressed the proliferation of CD4[+], but not CD8[+] T cells. In addition, NANA induced elevated levels of PD-1 expression in human CD4[+] T cells (Fig. 6d, e). To gain mechanistic insight into the transcriptional programs linking NANA to immune deregulation, we performed bulk RNA-sequencing of the same human T-cell cultures (Supplementary Fig. 14f). We found that NANA significantly upregulated 258 genes and downregulated 484 genes (DESeq2, FDR < 0.050; Methods; Fig. 6f; Supplementary Data 4). Upregulated gene set enrichment analysis highlighted several pathways associated with T-cell activity, such as *clathrin-dependent endocytosis*, *T-cell activation*, and *T-cell*

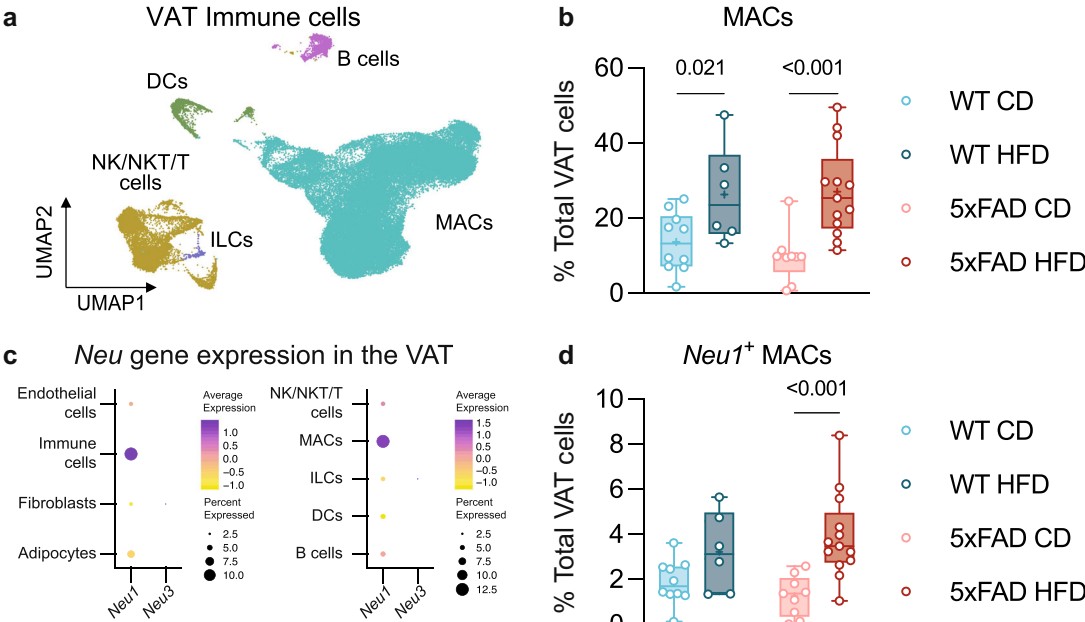

**Fig. 5 | *Neu1*-expressing macrophages in the mouse visceral adipose tissue are a potential source of NANA. a** Cellular landscape of the mouse visceral adipose tissue (VAT) immune cells across all genotype and diet conditions. UMAP embedding of single nuclei profiles (sNuc-Seq), colored after post hoc cell type annotation. Mice from three independent experiments, sample *n*: WT CD = 10, WT HFD = 6, 5xFAD CD = 9, 5xFAD HFD = 13. **b** HFD increased the frequency of VAT macrophages (MACs). Changes in frequency of the other VAT immune cell types are shown in Supplementary Fig. 11e. Annotations are as in Supplementary Fig. 11d, e. Sample *n*: see **a**. **c** Dot plots featuring the expression of sialidase *Neu* genes (color scale) and the percentage of cells expressing them (dot size) in the overall VAT (left) and VAT immune compartment (right). Of the four mammalian *Neu* genes, only *Neu1* and *Neu3* transcripts were detected. **d** HFD increased the frequency of *Neu1*-expressing macrophages in the VAT. Sample *n*: see (**a**). **b**, **d** Statistical analyses: two-way ANOVA followed by Fisher's LSD post hoc test. Box plots represent the minimum and maximum values (whiskers), the first and third quartiles (box boundaries), the median (box internal line), and the mean (cross). Source data are provided as a Source Data file.

*differentiation* (hypergeometric test, FDR < 0.050, Fig. 6g; Supplementary Data 5). Consistent with the results from T-cell proliferation assays, the top-downregulated pathways were found to be related to cell proliferation (Fig. 6g; Supplementary Data 5). Moreover, we found prominent suppression of genes involved in cell metabolic and bioenergetic pathways, and specifically *NAD metabolic process*, *biosynthesis of amino acids*, and *glycolysis/gluconeogenesis* (Fig. 6g; Supplementary Data 5), thus suggesting that NANA impacts T-cell function, at least in part, via disrupted cell metabolism.

To determine whether the increased levels of circulating NANA could contribute to immune rearrangements, we first tested the effects of NANA on the immune system in WT animals. To this end, we used both WT young-adult (6.5–9 mo) and middle-aged (11–14 mo) mice and treated them with repeated administrations of NANA, and subsequently analyzed their splenic T-cell compartment by flow cytometry one day after the last injection (Fig. 7a–c; Supplementary Fig. 15a). In young-adult mice, we did not observe any effect of NANA (CD4⁺ T cells, Fig. 7b; CD8⁺ T cells, Supplementary Fig. 15b). In contrast, treatment of middle-aged mice with NANA substantially recapitulated the CD4⁺ T-cell rearrangements observed in HFD-fed 5xFAD mice (Fig. 7c), whereas the effect on CD8⁺ T cells was marginal (Supplementary Fig. 15c). Therefore, it appears that NANA induced immune rearrangements in mice that are relatively susceptible to stress factors, as is the case with middle age and neurodegeneration[52].

The effect of NANA administration on the immune system of middle-aged WT mice encouraged us to test whether administration of NANA to 5xFAD mice would lead to an effect of HFD on NOR performance (Fig. 7d). Using the NOR test 3 weeks after one week of repeated injections of NANA, we found loss of recognition memory at the time when the age-matched PBS-injected 5xFAD control mice hardly showed any impairment as compared to the age-matched PBS-injected WT mice (Fig. 7e; Supplementary Fig. 16a–c). Right after behavior

assessment we euthanized the mice and analyzed the fate of their splenic CD4⁺ T-cell profile (Supplementary Fig. 16d). We found a significant reduction in naive cells and a significant elevation of TEMs in NANA-injected 5xFAD mice relative to PBS-injected 5xFAD controls (Fig. 7f). In one of the mouse cohorts, we also analyzed the profile of CD4⁺ T cells in the blood and found a stronger effect than in the spleen of the same mice, including a significant increase in circulating FOXP3⁺CD25⁺ Tregs and PD-1⁺ cells (Supplementary Fig. 16e). Of note, whereas the analyses in the WT mice described above were performed one day after the last NANA injection (Fig. 7a), here, in the 5xFAD mice, the immune profiling was performed 3-4 weeks after the last NANA injection (Fig. 7d). In conclusion, NANA administration could affect CD4⁺ T cells in vivo under conditions of reduced resilience, causing aging-like rearrangements compatible with terminal differentiation, immune-suppression, and metabolic dysfunction, and accelerating deterioration of novelty discrimination capability in neurodegeneration-prone animals.

## Discussion

In the present study, we found that a long-term obesogenic diet regimen accelerated disease manifestations in 5xFAD mice, with no effect on age-matched WT mice. We further show that the primary disease effect on the brain and adipose tissue were due to the neurodegenerative process and the diet, respectively. The comorbidity effect in HFD-fed 5xFAD mice was functionally linked to quantitative and qualitative changes in splenic CD4⁺ T cells and elevation of plasma levels of free NANA. In vitro, NANA was found to induce CD4⁺ T-cell deregulation, which was confirmed in independent in vivo experiments performed in mice with relatively low-resilience conditions (5xFAD mice and middle-aged WT mice), but not in young-adult WT mice.

There are several studies supporting the contention that conditions that dampen systemic immunity contribute to the accelerated

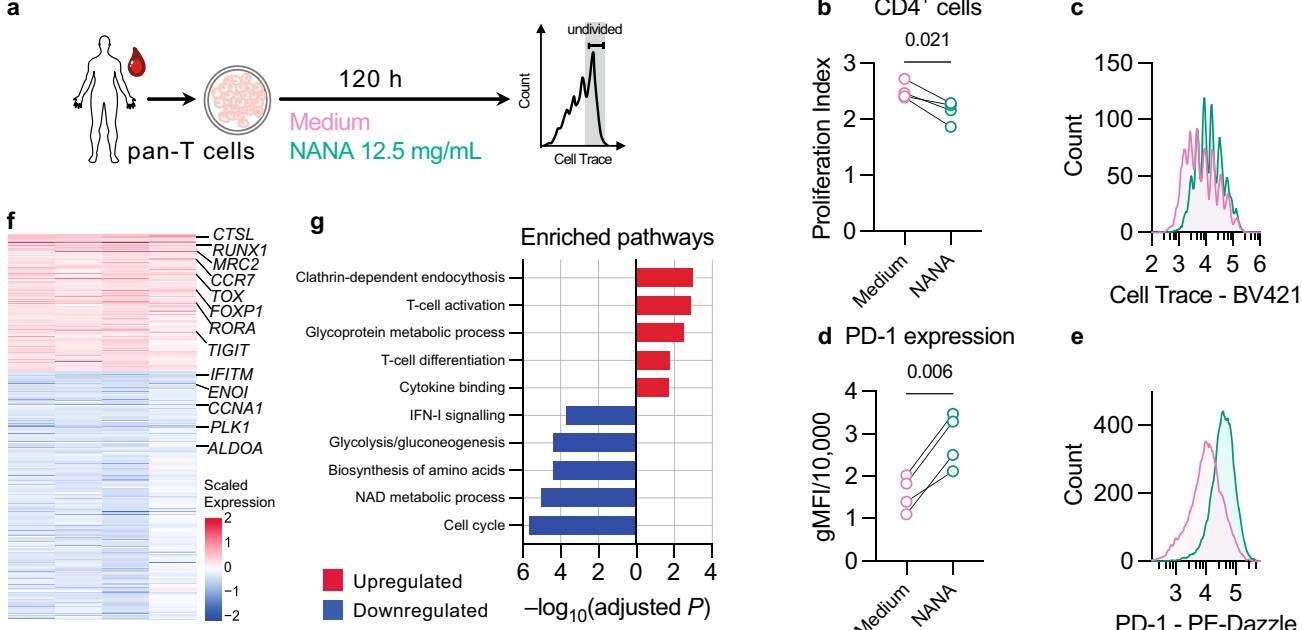

**Fig. 6 | NANA induced human T-cell exhaustion in vitro. a−e** Effect of NANA on human T cells from peripheral blood cultured in vitro. **a** Schematic presentation of treatment regimen. **b−e** Assessment of proliferative ability (**b**, **c**) and PD-1 geometric mean fluorescence intensity (gMFI; **d**, **e**) in human CD4⁺ T cells. The shown experiment is one of three independent experiments where different NANA concentrations were tested (Methods, *Data reporting* section). Sample *n* = 4 individuals. From each individual, one aliquot of T cells was treated with NANA, and one with medium as control. Statistical analyses: paired two-tailed Student's *t* test. **b**, **d** Black lines connect paired points. **c**, **e** Histograms of representative samples. **f**, **g** Bulk RNA-Seq of the same human T-cell cultures in (**a−e**). **f** Heatmap showing significantly (FDR-adjusted DESeq2 *P*-value <0.050) upregulated (red) and down-regulated genes (blue). Each column represents one individual and each row one gene. For each individual, gene expression values are expressed as log-transformed fold-changes (NANA versus medium control). **g** Bar plot showing significantly (FDR-adjusted hypergeometric test *P*-value <0.050) upregulated (red) and down-regulated (blue) pathways. **b−e**, **g** Source data are provided as a Source Data file.

progression of neurodegenerative diseases. Studies using immune-compromised mice have demonstrated that CD4⁺ T cells are specifically involved in multiple aspects of brain function, including microglia maturation[53], adult hippocampal neurogenesis[10], spatial navigation[54], and emotional behavior[55]. Here we found reduced naive cells and increased TEMs and FOXP3⁺ Tregs in the splenic CD4⁺ T-cell compartment of HFD-fed 5xFAD mice. These changes mirror those occurring with immune aging in mice and humans[45–47,56]. Furthermore, we found increased proportions of exhausted CD4⁺ TEMs. Overall, our results are in line with studies on peripheral blood from human AD patients showing reduced CD4⁺ naive T cells and increased CD4⁺ TEMs[57,58], and with the recent report of AD-associated increased PD-L1 expression in several circulating T effector-cell subsets, including CD4⁺ cells[59]. Findings on the proportions or the immune-suppressive capacity of circulating CD4⁺ Tregs in human AD patients are contradictory, with studies reporting increase, decrease, or no change[58,60–66]. Studies in AD mouse models demonstrated that homing of CD4⁺ Tregs to the brain is associated with disease amelioration[20,67]. However, the contribution of Tregs to withstanding chronic neurodegeneration might depend on their location (i.e. whether they are in the brain or in the periphery) and disease stage. This could explain the apparent contradiction of independent studies in AD mouse models showing, on the one hand, that early and repeated depletion of Tregs is detrimental[68,69], whereas on the other hand, late and transient depletion promotes their homing into the brain and disease amelioration[20]. It is therefore conceivable that the increase of CD4⁺ Tregs in HFD-fed 5xFAD mice might initially reflect the attempt to restrain peripheral inflammation due to the concomitant expansion of T effector cells[55,66]. However, in the long term, persistently high levels of CD4⁺ Tregs in the periphery might interfere with the brain's ability to recruit reparatory immune cells, such as bone marrow-derived myeloid cells as well as Tregs[7,20,67,70–73].

Upon metabolite profiling, we identified free NANA as the diet-associated metabolite in the circulation that displayed the strongest association with both the decline of recognition memory and the elevation of splenic Tregs levels in HFD-fed 5xFAD mice. Since we did not test non-polar metabolites, we cannot rule out the possibility that additional HFD-related metabolites may have effects on disease manifestations. Our data suggest that neuraminidase *Neu1*-expressing macrophages in the VAT may be a putative source of NANA in obesity. Our findings are in line with previous reports of increased NEU1 enzymatic activity in the VAT of two strains of obese mice[74] and attenuated weight gain and VAT inflammation in mice with diet-induced obesity treated with a pan-neuraminidase inhibitor[75]. *Neu1* expression is required during monocyte-to-macrophage differentiation and for macrophage activation and phagocytosis[76–78]. Therefore, accumulation of *Neu1*-expressing VAT macrophages in obesity may contribute to chronic local inflammation, which may in turn negatively affect systemic immunity. Although we cannot rule out the possibility that NANA had direct effects on the brain, for example via microglia or astrocytes, as previously reported for other metabolites[79,80], the amount of free NANA in the hippocampus was comparable across diet and genotype groups.

Overall, our data suggest that the effects of NANA on the immune cells in the periphery were the driving force of the accelerated disease manifestations in HFD-fed 5xFAD mice. Consistently, CD4⁺ T cells of both mice and humans were more susceptible to NANA than CD8⁺ T cells, in agreement with our finding that CD4⁺ T cells were specifically perturbed in HFD-fed 5xFAD mice. Furthermore, the effects of NANA on cultured human T cells, which encompass augmented expression of PD-1 protein and transcriptional changes associated with T-cell activation/differentiation, metabolic perturbations, and reduced cell proliferation, mirror our findings in HFD-fed 5xFAD mice of increased levels of CD4⁺ TEMs (i.e. differentiated cells) and expression of

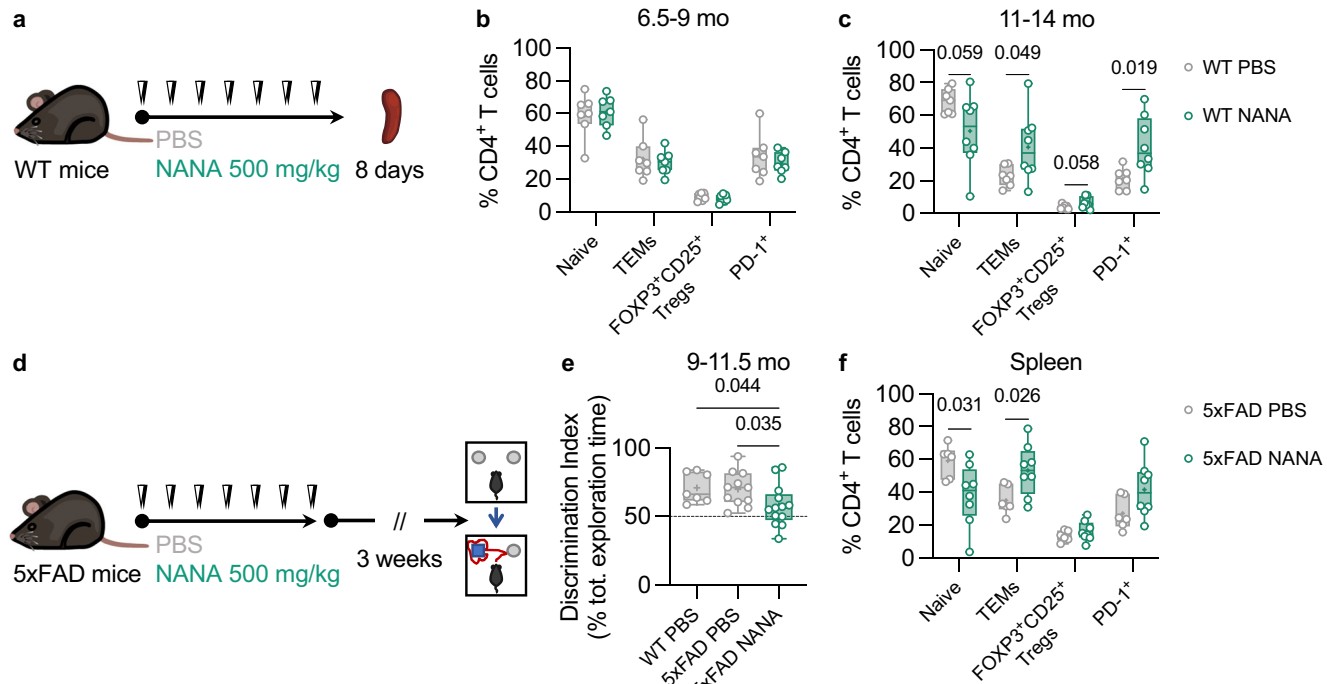

**Fig. 7 | NANA administration induced CD4+ T-cell deregulation in vivo and accelerated cognitive decline in 5xFAD mice. a–c** Effect of NANA administration on the spleen immune profile of WT mice. **a** Schematic presentation of treatment regimen. NANA or PBS control was subcutaneously injected twice a day, once in the morning (white arrowheads) and once in the evening (black arrowheads), for 7 consecutive days. **b**, **c** Flow cytometric quantification of splenic frequencies of CD4+ naive T cells (CD44lowCD62Lhigh), CD4+ TEMs (CD44highCD62Llow), CD4+FOXP3+CD25+ Tregs, and CD4+PD1+ T cells in young-adult (**b**) and middle-aged mice (**c**). For both (**b**) and (**c**), data from two independent experiments, sample *n*: **b**, PBS = 7, NANA = 7; **c**, PBS = 7, NANA = 8. Statistical analyses: **b**, multiple two-tailed unpaired Student's *t* tests; **c**, multiple two-tailed unpaired Student's *t* tests with Welch's correction (Methods, *Statistical analyses* section). **d–f** Effect of NANA on novelty discrimination and CD4+ T-cell profile in 5xFAD female mice. **d** Schematic

presentation of treatment regimen. 5xFAD female mice were treated with NANA or PBS control as in (**a**); PBS-injected age-matched WT female controls were also included. Three weeks after the last injection, novelty discrimination was assessed using the NOR test. **e** Results of the NOR test. Data from three independent cohorts, age at cognitive assessment: 9, 10, and 11.5 mo, sample *n*: WT PBS = 7, 5xFAD PBS = 11, 5xFAD NANA = 13. Statistical analyses: one-way ANOVA followed by Fisher's LSD post hoc test. **f** Spleen CD4+ T-cell profile. Data from two of the three cohorts described in (**e**), age at cognitive assessment: 9 and 11.5 mo, sample *n*: 5xFAD PBS = 6, 5xFAD NANA = 8. Statistical analyses: multiple two-tailed unpaired Student's *t* tests. **b**, **c**, **e**, **f** Box plots represent the minimum and maximum values (whiskers), the first and third quartiles (box boundaries), the median (box internal line), and the mean (cross). Source data are provided as a Source Data file.

exhaustion markers, including PD-1. Of note, the negative impact of NANA might be dependent on some predisposing condition by the host. In line, we found that the immune system of relatively aged WT mice and 5xFAD mice was more vulnerable to the effects of NANA, and that 5xFAD mice treated with NANA displayed accelerated loss of recognition memory.

Taking together our in vivo and in vitro immune profiling results, we suggest that the accelerated disease manifestations in HFD-fed 5xFAD mice are caused by the HFD-accelerated aging of the CD4+ T-cell compartment, at least in part driven by NANA. The aging-like changes in the CD4+ T-cell compartment impair CD4+ T-cell functionality in the periphery[56] and, thus, could contribute to the diminished ability of the immune system to support brain homeostasis and to facilitate coping with brain pathology[81]. In line with this contention are the recent studies demonstrating that aging of the immune system is sufficient to drive aging of non-lymphoid solid organs, including the brain[82], and is linked to cognitive decline[12]; the general anti-aging effects of systemically targeting the immune exhaustion-related molecule PD-1[83]; and several independent studies demonstrating both alleviated brain pathology and improved cognition in different mouse models of neurodegeneration after systemic blockade of the inhibitory immune checkpoints PD-1/PD-L1[21–24] and, more recently, ERMAP[25]. Of note, the lack of effect of PD-1 blockade on amyloid β plaques burden in the study of Latta-Mahieu et al. 2018[84] could be an outcome of the experimental conditions. Intriguingly, elevation of total sialic acid in the circulation has been

observed in a wide range of AD comorbidities, such as aging[85], obesity[86], diabetes[87] and cardiovascular disease[88], in addition to AD itself[89]. Thus, we propose that sialic acid-driven systemic immune deregulation may be a risk factor not only in obesity, but generally in pathological states undermining organismic resilience, and especially in compound AD-comorbid states. Therefore, strategies rejuvenating the immune system or therapies targeting diet-induced metabolites like NANA[75] may provide salutary benefits for dementia modification regardless of underlying etiology.

## Methods

### Mice

All experiments detailed herein complied with the regulations formulated by the Institutional Animal Care and Use Committee (IACUC) of the Weizmann Institute of Science (application numbers: 03960618-3, 01200121-2, 03230322-2). Female and male mice were bred and maintained by the Animal Breeding Center of the Weizmann Institute of Science. Housing conditions were: 12-hour dark/light cycle (lights on at 8 am), temperature 22 °C, humidity 30-70%. For comorbidity studies, heterozygous 5xFAD transgenic mice[33] (line Tg6799, The Jackson Laboratory) on a C57/BL6-SJL background and age-matched wild-type (WT) controls were used. Genotyping was performed by PCR analysis of ear clipping DNA, as previously described[33]. Since the C57/BL6-SJL strain carries the retinal degeneration *Pde6b^rd1* mutation, which causes visual impairment in homozygosis (https://www.jax.org/strain/100012), mice were further tested for presence of the allele, as

previously described[90]. To avoid gut microbiota-related cage effects due to coprophagia[91], 5xFAD and WT mice were housed together. For the study of the effects of NANA on the immune system in vivo, we used four cohorts of female and male WT mice, and specifically: three cohorts of C57/BL6-SJL mice, age 6.5, 9, and 14 mo; one cohort of C57/BL6 mice, age 11 mo. For the study of the effects of NANA on novelty discrimination, female C57/BL6-SJL 5xFAD and age-matched WT controls were used. To avoid NANA assimilation with coprophagia, NANA-injected mice were housed separately from the PBS-injected controls. All mice were provided with standard chow (calories from proteins: 24%; calories from carbohydrates: 58%; calories from fat: 18%; 2918, Teklad), placed on a hopper integrated with the cage lid, and water *ad libitum*, and housed in cages enriched with one paper shelter. For comorbidity studies, to induce obesity, at 6-9 weeks of age, mice were switched to a high-fat diet (HFD; calories from proteins: 18%; calories from carbohydrates: 22%; calories from fat: 60%; TD.06414, Teklad), and the food pellet checked twice a week for replenishment. Control mice were kept on standard chow (control diet, CD). Mice allocated for behavioral studies or NANA/PBS injections were switched to a 12-h reversed dark/light cycle (lights on at 8 pm) at least 7 days prior to behavior assessment, and maintained in the regimen until experimental endpoint.

### Glucose tolerance test

Prior to test, mice were fasted overnight for 16 h, and tail blood glucose measured for 0 time point (FreeStyle glucose meter and strips). Subsequently, mice were given 5 μL/g body weight of a 200 mg/mL glucose (Sigma-Aldrich)/water solution intraperitoneally, and blood glucose was measured at 15, 30, 60, 90, and 120 min after glucose injection.

### Novel Object Recognition (NOR) test

Mice that were homozygotes for the *Pde6b^rd1* allele, which causes visual impairment, were not used. A previously published protocol[92] was modified as follows. Mice were handled daily for at least 7 days prior to the initiation of NOR test. The NOR test spanned 2 days and included 3 trials: habituation trial, day 1 (20 min session in the empty arena); familiarization trial, day 2 (10 min session in the arena with two identical objects located in opposite corners of the floor, approximately 9 cm from the walls on each side); test trial, day 2, 60-70 min after familiarization (6 min session in the arena with one of the objects replaced by a novel one). Familiar and novel objects were visually and tactually distinct. To control for potential positional preference, the location of the novel object relative to that of the familiar object was randomized. Two identical arenas placed side by side were used simultaneously (one mouse in each arena). Each arena was a 41.5×41.5 cm gray plastic box. Mouse behavior was recorded and measured blindly using the EthoVision XT 11 automated tracking system (Noldus). Novel object preference was measured as discrimination index, expressed as time of interaction with the novel object relative to time of interaction with both objects (%) during the test trial. A discrimination index above 50% indicates novelty recognition, with 50% indicating no recognition. After each trial, the arenas and equipment were wiped with 10% ethanol. Female and male mice were tested on different days. For longitudinal assessment of cognitive performance (comorbidity studies), objects were changed; in addition, to control for potential arena effect, mice tested in one arena at 6.5 mo were tested in the other arena at 8 mo. Each mouse was presented two different novel objects at 6.5 and 8 mo.

### Exogenous NANA administration

Mice were subcutaneously injected with 20 μL/g body weight of a 25 mg/mL NANA (Biosynth-Carbosynth)/PBS solution (pH adjusted to 7.4) twice a day (8-9 h interval) for 7 consecutive days. Control mice were given PBS.

### Mouse blood and tissue collection and processing

In all experiments, sacrifices were carried out in the morning (from 9 am to 1 pm). Mice were anesthetized, exsanguinated by heart puncture, and transcardially perfused with PBS. For plasma metabolite analysis, blood was collected in tubes containing 20 μL heparin and spun at 3,000 g for 15 min at 4 °C. Supernatant plasma was aliquoted and snap frozen in liquid nitrogen (LN). For immune profiling, peripheral blood mononuclear cells (PBMCs) were isolated by density gradient centrifugation protocol (Cytiva™ – density 1.084 ± 0.001 g/ml). After dissection, the following brain regions were excised: for histology, left hemisphere, post-fixed in 4% paraformaldehyde (PFA)/PBS; for sNuc-Seq, left hippocampus, snap frozen in LN; for ELISA and biochemistry, right cortex and right hippocampus, snap frozen in LN. The spleen was mashed with the plunger of a syringe against a 70 mm strainer and treated with ammonium-chloride-potassium (ACK) lysis buffer (Gibco™) to remove erythrocytes. Splenocytes were then used immediately for flow cytometry and pan-T-cell cultures, while for CyTOF, aliquots were resuspended in cell freezing medium (Sigma-Aldrich) and frozen in a Mr. Frosty container (Thermo Fisher) at −80 °C. The left gonadal fat pad was snap frozen in LN for sNuc-Seq.

### Human samples

The study was approved by the Institutional Review Board of the Rambam and Galilee Medical Centers (application number: 0013-20-RMB/66756). All donors were informed on the purpose of the study and gave their consent. Blood was collected from four healthy volunteers (two male and two female subjects). All subjects, average 34 years, were free from acute infectious diseases and in good physical condition. PBMCs were freshly isolated from blood collected in EDTA-coated vials by layering diluted blood (1:1 in PBS) on top of an equal volume of Ficoll (GE Healthcare Life Sciences™ – density 1.077 ± 0.001 g/ml), followed by centrifugation and isolation of the buffy coat.

### Flow cytometry

Splenocytes or PBMCs were washed with ice-cold PBS and stained with LIVE/DEAD™ Fixable Aqua Dead (Thermo Fisher) according to the manufacturer's instructions. After fragment crystallizable (Fc) blocking (Biolegend), cells were stained for surface antigens. For intranuclear staining, a FOXP3 staining kit was used (Invitrogen) according to the manufacturer's instructions. The antibodies used, preconjugated to fluorophores, are reported in Supplementary Table 1. To assess apoptosis of human lymphocytes following in vitro exposure to NANA, the FITC Annexin V Apoptosis Detection Kit with PI (Biolegend) was used, according to the manufacturer's instructions. Flow cytometry data were acquired with CytExpert on a CytoFLEX S system (Beckman Coulter) and analyzed using FlowJo software (v10; Tree Star). In each experiment, relevant negative, single-stained, and fluorescence-minus-one controls were used to identify the populations of interest.

### Lymphocyte cultures

Pan-T cells from mouse splenocytes and human PBMCs were isolated by negative selection using magnetic beads (Miltenyi) according to the manufacturer's instructions. From each mouse or human individual, isolated cell aliquots were plated in at least duplicates for each treatment condition, except in the experiment presented in Supplementary Fig. 13c, where single technical replicates were used. Isolated cells were stained with CellTrace Violet Cell Proliferation Kit (Thermo Fisher) according to the manufacturer's instructions, and 8×10^4 cells were plated into 96-well U-bottom and activated with anti-CD3/anti-CD28-coated Dynabeads (Thermo Fisher), as previously described[93]. Cells were then cultured for 96 h (mouse cell cultures) or 120 h (human cell cultures) in RPMI (BioSource International) supplemented with 5% fetal bovine serum, 5 mM glutamine, 25 mM Hepes (Sigma-Aldrich),

and 1% antibiotics (Invitrogen), with or without NANA (pH adjusted to 7.4). Samples were acquired on CytoFLEX S (Beckman Coulter) and analyzed using FlowJo. To assess proliferative ability, the Proliferation Tool of FlowJo was used to estimate the proliferation index, i.e. the total number of cell divisions divided by the number of cells that underwent division.

## Mass cytometry

For intracellular staining, short-term reactivation of cryopreserved splenocytes and subsequent mass cytometry analysis were performed as described previously[94]. In short, cells were kept at −80 °C for less than 2 months before thawing in a 37 °C water bath. Cells were then immediately resuspended in cell culture medium supplemented with 1:10,000 benzonase (Sigma-Aldrich), and centrifuged at 300 g for 7 min at 24 °C. Samples were then left overnight at 37 °C before restimulation with 50 ng/mL phorbol 12-myristate 13-acetate (Sigma-Aldrich) and 500 ng/mL ionomycin (Sigma-Aldrich) in the presence of 1× brefeldin A (BD Biosciences) and 1× monensin (Biolegend) for 4 h at 37 °C. For splenocytes, one anchor sample to correct batch effect among different acquisitions and one non-stimulated control sample were also included. For both PBMCs and reactivated cryopreserved splenocytes, surface staining was performed for 30 min at 4 °C. To identify dead cells, 2.5 μM cisplatin (Sigma-Aldrich) was added for 5 min on ice. To minimize technical variability, an equal number of cells from each sample were barcoded using Cell-ID 20-Plex (Fluidigm). Cells from all samples were then combined in one single tube. The combined sample was then processed for Live/Dead and surface staining. For intracellular cytokine staining of reactivated cryopreserved splenocytes, after surface staining, cells were fixed and permeabilized with FOXP3 staining kit (Invitrogen) and stained for intracellular markers and cytokines. The antibodies used are reported in Supplementary Table 2. Because CD4 molecules are internalized during phorbol 12-myristate 13-acetate/ionomycin stimulation[95], the anti-CD4 antibody was used in both the surface staining and the intracellular staining steps. After washing, the combined sample was incubated with 4% PFA overnight at 4 °C. Prior to acquisition in a Helios™ II CyTOF® system, samples were washed with cell staining buffer and mass cytometry grade water. Multidimensional datasets were analyzed using FlowJo and R (R Core Team, 2017).

**Acquisition and data pre-processing.** Quality control and tuning of the Helios™ II CyTOF® system (Fluidigm) was performed on a daily basis and before each acquisition. Samples were acquired in two separate days, and data were normalized using five-element beads (Fluidigm) that were added to the sample immediately before every acquisition, and using an anchor sample included in each reading, as previously described[96]. For analysis, live single cells were identified based on event length, DNA ($^{191}$Ir and $^{193}$Ir) and live cell ($^{195}$Pt) channels using FlowJo. Samples were then debarcoded using Boolean gating and mass cytometry data were transformed with an inverse hyperbolic sine (arsinh) function with cofactor 5 using the R environment.

**Algorithm-based high-dimensional analysis.** After gating for CD4$^+$ T cells using FlowJo, pre-processed data were considered for the analysis. All FlowSOM-based k-NN clustering was performed on the combined dataset to enable identification of small populations. For CD4$^+$ TEMs, resulting nodes were meta-clustered with the indicated k-values (based on the elbow criterion) and annotated manually. FlowSOM k-NN clustering and two-dimensions UMAP projections were calculated using the CyTOF workflow package (v. 1.2)[97].

## Plasma preparation for metabolite profiling

Plasma samples from some of the male mice included in the experiments described in Fig. 1c and in Fig. 3a–c were used. Extraction and analysis of polar metabolites were performed as previously

described[98,99], with some modifications: 100 μL of plasma were mixed with 1 mL of a pre-cooled (−20 °C) homogenous methanol (MetOH):methyl-tert-butyl-ether (MTBE) 1:3 (v/v) mixture. The tubes were vortexed and then sonicated for 30 min in ice-cold sonication bath (taken for a brief vortex every 10 min). Then, UPLC-grade water (DDW):MetOH (3:1, v/v) solution (0.5 mL) containing internal standards: $^{13}$C- and $^{15}$N-labeled amino acids standard mix (Sigma-Aldrich) was added to the tubes followed by centrifugation. The upper organic phase was transferred into 2 mL Eppendorf tube. The polar phase was re-extracted as described above, with 0.5 mL of MTBE. Both organic phases were combined and dried in speedvac and then stored at −80 °C until analysis. Lower, polar phase used for polar metabolite analysis was lyophilized, dissolved in 150 μL of 1:1 DDW:MetOH (v/v), centrifuged at 20,000 g for 5 min, transferred to a new tube, and centrifuged again. For analysis, 80 μL were transferred into HPLC vials.

## Human Aβ1-42 ELISA

Samples were homogenized in Tris Buffered Saline EDTA (TBSE) solution (50 mM Tris, 150 mM NaCl, and 2 mM EDTA, pH 7.4) with the addition of 1% Protease Inhibitor Cocktail (Sigma-Aldrich) using a microtube homogenizer with plastic pestles (for hippocampus; 1 mL/100 mg tissue) or a glass homogenizer (for cortex; 1 mL/200 mg tissue). The homogenates were then centrifuged for 40 min at 350,000 g in 500 μL polycarbonate centrifuge tubes (Beckman Coulter) at 4 °C in an Optima MAX-XP Ultracentrifuge with a TLA 120.1 rotor (Beckman Coulter). The supernatant (TBSE-soluble fraction) was collected, aliquoted, and stored at −80 °C until use. BCA assay (Pierce BCA Protein Assay Kit) was performed to determine total protein amount for normalization. To quantify Aβ1-42 peptides, the human Aβ42 Ultrasensitive ELISA Kit (Invitrogen) was used according to the manufacturer's instructions. Data were acquired using a Spark microplate reader (Tecan).

## LC-MS polar metabolite analysis

Metabolic profiling of polar phase was done as previously described[99] with minor modifications described below. Briefly, analysis was performed using Acquity I class UPLC System combined with mass spectrometer Q Exactive Plus Orbitrap™ (Thermo Fisher) that was operated in a negative ionization mode. The LC separation was done using the SeQuant Zic-pHilic (150×2.1 mm) with the SeQuant guard column (20 × 2.1 mm; Merck). The composition of mobile phase B and A was acetonitrile, and 20 mM ammonium carbonate with 0.1% ammonium hydroxide in DDW:acetonitrile (80:20, v/v), respectively. The flow rate was kept at 200 μL/min and gradient as follows: 0–2 min 75% of B, 14 min 25% of B, 18 min 25% of B, 19 min 75% of B, for 4 min.

**Polar metabolite data analysis.** The data processing was done using TraceFinder (Thermo Fisher). The detected compounds were identified by accurate mass, retention time, isotope pattern, fragments, and verified using in-house-generated mass spectra library. Peaks were quantified by calculating the area under curve (AUC) and then normalizing the AUC values by internal standards and original sample volume. A total of 229 metabolites were identified.

**Metabolite selection for follow-up studies.** For each of the identified metabolite, a linear regression model was built with metabolite plasma levels (scaled values) across all samples of all experimental conditions as the dependent variable, and the interaction between the two categorical "genotype" and "diet" variables ("genotype:diet" term) as the only predictor. To avoid collinearity, the intercept was set to 0. The overall approach is equivalent to a Cell Means Model, which fits an individual mean for each of the predictor levels. The metabolites whose models had unadjusted one-way *omnibus* ANOVA test *P*-value <0.050 and *P*-value for the 5xFAD:HFD condition <0.050 were considered "metabolites of interest". The metabolites of interest were, in

total, 22. For each metabolite of interest, the correlation between the metabolite's levels and the NOR discrimination index or splenic Tregs abundance across all samples of all genotype:diet conditions was determined using the Spearman's rank correlation. For each metabolite of interest, the coefficients of the Spearman's rank correlations ($\rho$) between metabolite level and NOR discrimination index (DI), and between metabolite level and splenic Tregs abundance, across all samples of all genotype:diet combinations, were calculated and plotted against each other ($\rho$-$\rho$ plot). The statistics related to the Cell Means Model and correlations, as well as the pairwise comparisons of each metabolite's abundance across genotype and diet groups, are reported in Supplementary Data 3.

## Measurement of NANA in the hippocampus

NANA was measured in the TBSE-soluble fraction (see *Human Aβ1-42 ELISA* section) using liquid chromatography with tandem mass spectrometry (LC-MS/MS).

**Materials.** Acetonitrile and formic acid of ULC/MS grade were from Bio-Lab (Israel). Water with resistivity 18.2 MΩ was obtained using Direct 3-Q UV system (Millipore). *N*-acetylneuraminic acid standard (NANA) was purchased from Biosynth-Carbosynth. $^{13}C_3$-*N*-acetylneuraminic acid ($^{13}C_3$-NANA) from Omicron Biochemicals was used as internal standard.

**Sample preparation.** To 100 µL of TBSE-soluble fraction from mouse hippocampi, 250 µL of ethanol and 10 µL of 10 µg/mL of internal standard were added, and the mixture was incubated in shaker (10 °C, 1,500 rpm, 3 h). The extracts were then centrifuged at 20,000 $g$ for 20 min. The obtained supernatants were evaporated, then resuspended in 100 µL of 20%-aqueous acetonitrile. For LC-MS/MS analysis, the samples were placed in 0.2-µm PTFE-filter vials (Thomson).

**LC-MS analysis.** The LC-MS/MS instrument consisted of Acquity I-class UPLC system (Waters) and Xevo TQ-S triple quadrupole mass spectrometer (Waters) was used for the analysis. MassLynx and TargetLynx software (v.4.1, Waters) were applied for the acquisition and analysis of data. Chromatographic separation was done on a 100 ×2.1-mm i.d. 1.7-µm UPLC Atlantis Premier BEH C18 AX column (Waters) with 0.2% formic acid as mobile phase A and 0.2% formic acid in acetonitrile as B at a flow rate of 0.3 mL/min and column temperature 25 °C. A gradient was as follows: 0.5 min the column was hold at 4% B, then linear increase from 1 to 15% B in 1 min, then to 40% B in 3 min, and to 100% B in 0.5 min, and hold at 100% B for 1 min. Then back to 1% B during 0.5 min, and equilibration for 1 min. Samples kept at 8 °C were automatically injected in a volume of 1 µl. The mass spectrometer equipped with an electrospray ion source and operated in negative ion mode was used, with 0.10 mL/min of argon as a collision gas flow. The capillary voltage was set to 1.85 kV, cone voltage 30 V, source offset 12 V, source temperature 120 °C, desolvation temperature 500 °C, desolvation gas flow 600 L/hr, cone gas flow 150 L/hr. Analytes were detected using corresponding multiple reaction monitoring (MRM): for NANA, 308.1 > 170.1 and 308.1 > 87.1 m/z, with collision energies 14 and 15 eV, respectively, and for $^{13}C_3$-NANA 311.0 > 173.0 and 311.0 > 90.0 m/z with collision energy 30 eV. The retention time of *N*-Acetylneuraminic acid peak was at 2.03 min. Sample NANA concentrations were calculated from the standard curve of 1–10,000 ng/mL and normalized by total protein amount (see *Human Aβ1-42 ELISA* section).

## Fluorometric assays on plasma samples

Assays were performed using commercially available kits (Abcam) according to the manufacturer's instructions, but scaling reaction volumes 1:5 to allow running in 384-well flat-bottom black plates (Greiner). To measure total cholesterol, the cholesterol assay kit – HDL and LDL/VLDL (ab65390) was used. To measure free NANA, the sialic acid (NANA) assay kit (ab83375) was used, and reactions, including background controls, were run for 1 h at RT. Samples were run in duplicates. Data were acquired using an Infinite 200 PRO microplate reader (Tecan) or a Spark microplate reader (Tecan).

## Measurement of plasma leptin

The mouse leptin ELISA kit (Abcam, ab100718) was used according to the manufacturer's instructions. Samples were run in duplicates. Data were acquired using an Infinite 200 PRO microplate reader (Tecan).

## Nuclei isolation and single-nucleus RNA library preparation

**Brain samples.** Hippocampus tissue specimens were kept frozen at −80 °C until processing. Samples were batched in sets of four representing all experimental and sex groups. Working on ice throughout the nuclei isolation process, the frozen hippocampus tissue was transferred into a Dounce homogenizer (Sigma-Aldrich, D8938) with 2 mL of lysis buffer. Lysis buffer used was either EZ Lysis Buffer, as we previously used[100] (Sigma-Aldrich, NUC101), or Igepal Lysis Buffer, containing 0.1% IGEPAL® CA-630 (Sigma-Aldrich, I8896), 10 mM Tris HCL pH 7.5, 146 mM NaCl, 3 mM MgCl₂, 40 U/mL of RNAse inhibitor (NEB M0314L, as we previously used[37]). Tissue was gently homogenized while on ice 15 times with pestle A followed by 15 times with pestle B, then transferred to a 15 mL conical tube. A further 3 mL of lysis buffer was added to a final volume of 5 mL, left on ice for 5 min, and then centrifuged in a swing bucket rotor at 500 $g$ for 5 min at 4 °C. Samples were processed two at a time, the supernatant was removed, and the pellets were left on ice while processing the remaining tissues to complete a batch of 4 samples. The nuclei pellets were then resuspended in a wash buffer containing 0.02% BSA (NEB B9000S) and 40 U/mL of RNAse inhibitor in PBS, and the volume adjusted to 5 mL by adding more wash buffer. The nuclei were centrifuged in a swing bucket rotor at 500 g for 5 mins at 4 °C. The supernatant was removed and the pellet was gently resuspended in 500 µL of wash buffer. Nuclei were filtered through a 30 µm MACS Smartstrainer (Miltenyi, 130-098-458) and counted using the LUNA-FL™ Dual Fluorescence Cell Counter (Logos Biosystems) after staining with Acridine Orange/Propidium Iodide Stain (Logos Biosystems, F23001) to differentiate between nuclei and cell debris. Sixteen thousand (16,000) nuclei were run on the 10x Single Cell RNA-Seq Platform using the Chromium Single Cell 3' Reagent Kits v3. Libraries were made following the manufacturer's protocol. Briefly, single nuclei were partitioned into nanoliter-scale Gel Bead-In-Emulsion (GEMs) in the Chromium controller instrument, where cDNA shares a common 10x barcode from the bead. Amplified cDNA was measured by Qubit HS DNA assay (Thermo Fisher, Q32851) and quality assessed by High Sensitivity D5000 ScreenTape (5067-5592) with High Sensitivity D5000 Reagents (5067- 5593) on the 2200 TapeStation system (Agilent). The WTA (whole transcriptome amplified) material was diluted to <8 ng/mL and processed through v3 library construction according to the manufacturer's protocol, and resulting libraries were quantified again by Qubit and TapeStation. Libraries from 4 channels were pooled and sequenced on 1 lane of NextSeq 550 (or 8 channels sequenced on two NextSeq 550 runs) at the Center for Genomic Technologies in the Institute of Life Sciences at The Hebrew University of Jerusalem, for a target coverage of around 150 million reads per channel.

**Visceral adipose tissue samples.** Visceral adipose tissue (VAT) specimens were kept frozen at −80 °C until processing. Nuclei were isolated with a previously published protocol using salt-Tris (ST)-based buffers[101]. Frozen VAT was placed into a gentleMACS™ C tube (Miltenyi, 130-096-334) with 1 mL TST buffer and homogenized using the gentleMACS™ Octo dissociator (Miltenyi). The sample was removed and, with a further 1 mL of TST, transferred to a 15 mL conical tube on

ice for 10 min. Next, the sample was filtered through a 40 µm Falcon™ cell strainer (Thermo Fisher, 08-771-1), to which 3 mL of 1x ST buffer was also added through the filter. The resulting 5 mL sample was then centrifuged at 500 g for 5 min at 4 °C in a swinging bucket centrifuge, following which the supernatant was removed and the pellet resuspended in 1x ST buffer (volume determined by pellet size). The nuclei-containing solution was transferred to a 5 mL polystyrene tube through a 35 µm cell strainer cap (Thermo Fisher, 08-771-23), and nuclei were counted using a C-chip disposable hemocytometer (VWR, 82030-468). Eight thousand (8,000) single nuclei were loaded into each channel of the Chromium single cell 3′ chip, for V3 10x technology (10x Genomics). Single nuclei were partitioned into GEMs and incubated to generate barcoded cDNA by reverse transcription. Barcoded cDNA was next amplified by PCR prior to library construction. Libraries of paired-end constructs were generated using fragmentation, sample index and adaptor ligation, and PCR was run according to the manufacturer's recommendations (10x Genomics). Libraries from four 10x channels were pooled together and sequenced on one lane of an Illumina HiSeq X (Illumina) by the Genomics Platform of the Broad Institute.

### Quality controls for sequencing and pre-processing of sNuc-Seq data

De-multiplexing of samples after Illumina sequencing was done using 10x Cellranger version 5.0.0 mkfastq to generate a Fastq file for each sample. Alignment to the mm10 transcriptome and unique molecular identifier (UMI)-collapsing were performed using the Cellranger count (version 5.0.0, mm10-2020-A_premrna transcriptome, single cell 3′ chemistry). Separate Fastq files of the same mouse sample were combined by running the CellRanger count with multiple fastqs input parameters. Since nuclear RNA includes roughly equal proportions of intronic and exonic reads, we built and aligned reads to a genome reference with pre-mRNA annotations, which account for both exons and introns.

**Technical artifacts and ambient RNA correction.** To account for technical artifacts in the data, specifically correcting gene counts shifted due to ambient RNA, we ran the CellBender[102] (version 2) program on each sample, which removes counts due to ambient RNA molecules and random barcode swapping from (raw) UMI-based single cell or nucleus RNA-Seq count matrices, and also determines which cell barcodes are valid nuclei libraries, excluding empty droplets and low-quality libraries. We used the CellBender output as input to downstream analysis.

**Data normalization.** For every nucleus, we quantified the number of genes for which at least one read was mapped, and then excluded all nuclei with fewer than 100 detected genes. For visceral adipose tissue (VAT) data, nuclei with more than 5000 genes were also filtered out. Genes that were detected in fewer than 3 nuclei were excluded. Expression values $E_{i,j}$ for gene $i$ in cell $j$ were calculated by dividing UMI counts for gene $i$ by the sum of the UMI counts in nucleus $j$, to normalize for differences in coverage, and then multiplying by 10,000 to create TPM-like values, and finally computing $\log_2(\text{TP10K}+1)$ using the *NormalizeData* function from the *Seurat* package[103–106] (version 4).

**Doublet detection.** We annotated each nucleus with a doublet score – the nucleus' probability of being a doublet, related to the fraction of artificially generated doublet neighbors (using an in-house optimization of DoubletFinder[107] with the following parameters: PCs = 1-45, pN = 0.25, pK = 150/(#cells), pANN = False and sct = False). This score would later be considered for the removal of doublets. We first used a high-resolution clustering (1.3 for the hippocampus and 1.5 for the VAT, see description under *Dimensionality reduction and clustering*). We excluded clusters that had more than 50% of cells that had over a high

doublet score (0.35 for the hippocampus and 0.4 for the adipose tissue). Second, cells from other clusters that had over a high doublet score were excluded. In the VAT: 10,625 doublets were removed and 275,336 nuclei remained in the dataset. In the hippocampus: we excluded from the analysis cells from clusters classified as endothelial cells or OPCs, since the doublet detection failed for these cell types. For OPCs, these specifically include cells differentiating from oligodendrocytes. At the end of this stage in the hippocampus dataset, 38,060 doublets were removed and 269,503 nuclei remained in the data set (for $n = 28$ mice, across all mouse genotype and diet groups). The downstream analysis of sub-clustering of specific cell types included a second inspection for doublets.

### Integrating datasets

**Identification of variable genes and scaling the data matrix.** After data pre-processing, samples from the four conditions within each batch were merged into a single Seurat object. For each batch, variable genes were selected by using a variance-stabilizing transformation (using FindVariableFeatures method, Seurat). This method, first, fits a line to the relationship of log(variance) and log(mean) of each gene from the non-normalized data, using local polynomial regression (LOESS); next, it standardizes the feature values using the observed mean and expected variance (given by the fitted line). Feature variance is then calculated on the standardized values and the genes with the top values were selected as variable genes for downstream analysis (2,500 for the hippocampus and 2,000 for the adipose tissue). For sub-clustering analysis of macrophages in the adipose tissue, genes in the ambient RNA signature were removed from the variable gene list to prevent clustering based on ambient RNA expression. The data were scaled, yielding the relative expression of each variable gene by scaling and centering (using ScaleData method, Seurat).

**Data integration.** After the identification of variable genes per batch, the batches (7 for the hippocampus and 2 for the VAT) were integrated into a single Seurat object (using Seurat v.4 integration workflow[103,104]), based on the CCA algorithm, using the methods FindIntegrationAnchors followed by IntegrateData. The integrated data matrix was then used for dimensionality reduction and clustering.

### Dimensionality reduction, clustering, and visualization

The integrated data matrix (restricted to the genes chosen as integration anchors) was then used for dimensionality reduction, visualization and clustering. Dimensionality reduction was done with principal component analysis (PCA, using RunPCA method Seurat). After PCA, significant principal components (PCs) were identified using the elbow method, plotting the distribution of standard deviation of each PC (*ElbowPlot* in Seurat). In the VAT analysis: 30 PCs were used. In the hippocampus analysis: 45 PCs for analysis of all cells, 20 PCs for astrocytes, 20 for microglia, and 10 for oligodendrocytes. Within the top PC space, transcriptionally similar nuclei were clustered together using a graph-based clustering approach. First, a k-nearest neighbor (k-NN) graph is constructed based on the Euclidean distance. For any two nuclei, edge weights were refined by the shared overlap of the local neighborhoods using Jaccard similarity (FindNeighbors method Seurat, with k = 60). Next, nuclei were clustered using the Louvain algorithm[108] which iteratively grouped nuclei and located communities in the input k-NN graph (FindClusters method Seurat, with resolution 0.5). Note that for the doublet detection stage on all cell types, we first used 45 PCs with a higher resolution clustering of 1.3 on data matrices that were merged based on the batch (see *Doublet detection* section). The obtained clusters were hierarchically clustered and re-ordered (using BuildClusterTree method Seurat). For visualization, the dimensionality of the datasets was further reduced by UMAP, using the same top principal components as input to the algorithm (using the RunUMAP method Seurat). Note that the

distribution of samples within each cluster was examined to eliminate that clusters were driven by batch or other technical effects. Clusters with low-quality cells (low number of genes detected, and missing or low-key cell-type marker genes and house-keeping genes such as *Malat1*), doublet clusters expressing markers of multiple cell types, and neuronal clusters from neighboring region of the hippocampal subiculum that appeared in an uneven form across samples, were removed from the analysis, leaving the hippocampus dataset with 237,631 (for $n = 28$ mice, across all mouse genotype and diet groups). Data visualization using UMAP showed that the clusters displayed a mixture of nuclei from all technical and biological replicates, with a variable number of genes, meaning the clustering was not driven by a technical effect.

**Sub-clustering analysis of cell types.** Specific cell types (i.e. microglia, astrocytes, oligodendrocytes, DG neurons, macrophages in the adipose tissue) were subsetted from the main dataset for a high-resolution analysis. For each such subset another cycle of clean-up was performed, removing doublet clusters based on different thresholds. Cells were clustered in high-resolution and clusters were then annotated and merged based on marker expression.

### Identification of clusters' cell types

Identification of cell types was done in the hippocampus using an in-house modification of a logistic regression model (linear_model.LogisticRegression from Python's sklearn package). The modifications included calculating the classification probability of each cell, and eventually associating the cell with the original cluster, and classifying the cluster as the overall highest scoring cell type. The classifier was trained on 80% of the nuclei (~50,000 nuclei) of our previously published and annotated sNuq-Seq data of the mouse hippocampus[37], and tested on the remaining 20% of the nuclei. Cell types were annotated according to the classification and further validated using known marker gene and as previously published[37]. In the VAT, identification of cell types was done based on known marker genes and the Bioconductor package SingleR[109].

### Cell fraction estimations and statistics

The fraction of different cell populations (i.e. clusters) was separately computed, for each sample across all clusters, as the fraction of nuclei in each cluster out of the total number of nuclei, by a parameter of interest (e.g. diet, sex, and more). Correlations between these fractions of interest were calculated using Spearman's rank correlation coefficient (with cor function from R's base package, method = "spearman"). For quantifications and statistical analyses, 219,237 nuclei were included from $n = 26$ samples: WT CD = 6, WT HFD = 7, 5xFAD CD = 6, 5xFAD HFD = 7. Two mice of the original $n = 28$ samples were excluded from the statistics due to technical artifacts that hampered their annotation to a specific member of one of the experimental groups (mice identity numbers: 636 and 640). To assess whether there was a significant change between experimental groups across conditions (diet, genotype), two-way ANOVA followed by Fisher's Least Significant Difference (LSD) post hoc test was used (see *Statistical analyses* section).

### Marker genes and differential expression analysis

Marker genes for glial and dentate gyrus (DG) granule neurons subclusters were found using the MAST test[110] (using batch and sex as latent variables for glial cells in the brain to account for technical effects), which was run using the FindMarkers function in Seurat v.4 (using the assay set to *RNA* to use the normalized UMI counts values per gene). To find diet-dependent differentially expressed signatures, we used the same scheme as above using the MAST algorithms with batch and sex as latent variables, within each glial cell type, comparing all cells of HFD-fed and CD-fed 5xFAD mice; similarly, WT mice were compared between diet groups. *P*-values were adjusted for multiple

hypothesis testing using Benjamini–Hochberg's correction (FDR). Adjusted *P*-value threshold of 0.010 was used to report significant changes and a fold change threshold of 0.250.

### Bulk RNA-Seq library preparation and analysis

Cultured human T cells after treatment were flash frozen in Buffer RLT (Qiagen, 79216) and kept in −80 °C until processing. Total RNA was extracted with the NucleoSpin RNA kit (Macherey-Nagel, 740955). One microgram (1 μg) total RNA was used as an input for mRNA isolation (NEB E7490S), followed by library preparation using NEBNext® Ultra™ II Directional RNA Library Prep Kit for Illumina (NEB E7760). Quantification of the libraries was done by Qubit and TapeStation. Paired-end sequencing of the libraries was performed on Nextseq 550. Demultiplexing of samples was done with Illumina's bcl2fastq software. The fastq files were next aligned to the human genome (hg38) using STAR[111] and the transcriptome alignment and gene counts were obtained with HTseq[112]. Aligned gene counts were normalized to fragments per kilobase of transcript per million mapped reads (log10(FPKM)) using the fpkm function in DESeq2 R package[113] (1.32.0). Genes with low counts (<10) were filtered before performing statistical analysis. Differential expression analysis of bulk RNA-Seq data between NANA-treated samples and control was performed using the DESeq2 R package (*P*-value adjusted to multiple hypothesis testing ≤0.050), while accounting for inter-sample differences. Significantly upregulated genes (adjusted *P*-value <0.050, log$_2$-fold change >0.250) and downregulated genes (adjusted *P*-value <0.050, log$_2$-fold change < −0.250) were determined. Up- and downregulated gene lists were separately functionally annotated with gene sets defined by the KEGG and Gene Ontology databases (org.Hs.eg.db, version 3.5.0), using enrichGO and enrichKEGG functions in R package *clusterProfiler* (3.6.0), using the hypergeometric *P*-value and FDR correction for multiple hypothesis (with a threshold of FDR < 0.050).

### Histology and immunohistochemistry

Paraffin-embedded tissue was sectioned with a thickness of 6 μm. One slide per animal was used for staining, each containing 5 equally spaced sections. Cresyl violet staining was performed to visualize neurons[22]. Antibody staining was performed as previously described[114], except that all primary antibodies were incubated overnight at RT, followed by another overnight at 4 °C. For Aβ staining, the Mouse On Mouse detection kit (Vector labs) was used according to manufacturer's instructions. The following primary antibodies were used: mouse anti-human Aβ (1:150; Covance); chicken anti-GFAP (1:150; Abcam). Cy2/Cy3-conjugated anti-mouse/chicken secondary antibodies (1:150; Jackson Immunoresearch) were used. For counterstaining, 4',6-diamidino-2-phenylindole (1:5000; Biolegend) was used.

### Microscopic imaging and analysis

Images were acquired using a fluorescence microscope (E800, Nikon) equipped with a digital camera (DXM 1200 F, Nikon), and with a ×20 NA 0.50 objective lens (Plan Fluor, Nikon). Quantitative analyses were performed by an experimenter blind to the identity of the animals, and using either the Image-Pro Plus software (Media Cybernetics) or ImageJ (NIH). Neuronal survival on cresyl violet-stained sections and Aβ plaques quantification were performed as previously described[22]. GFAP intensity was measured using the ImageJ software by applying a segmentation algorithm to mask stained areas (Otsu's method) and subsequently measuring average integrated density over 3-5 sections per animal. For each animal, stained sections' quantified values were averaged. Representative images were optimized using ImageJ and processed equally for all experimental conditions displayed.

### Experimental design

No statistical method was used to predetermine sample sizes, which were chosen with adequate statistical power based on the literature

and past experience[21,22,67,115]. The specific sample sizes and tests used to analyze each set of experiments are indicated in the Figure legends. Animals were randomly allocated to experimental groups balancing sex and genotype. Sample selection for subsequent analyses such as CyTOF, sNuc-Seq, and metabolite profiling was based on behavioral (NOR test), metabolic, flow-cytometric, and/or histological analyses. For in vitro experiments with both mouse and human lymphocytes, subjects were not divided in groups, but from each subject (mouse or human) one aliquot of cells was treated with NANA and another one with medium as control. Investigators were blind to animal identity during experiments and outcome assessment, except during behavior experiments in comorbidity studies, where diet groups, but not genotypes, were obvious. For in vitro experiments with both mouse and human lymphocytes, blinding was not relevant, as data were acquired under the same conditions using the software described in the Methods.

### Data reporting
Mice of both sexes were included in all experiments, except in the following cases: only male mice were used for plasma metabolite profiling (Fig. 4a–d and Supplementary Fig. 10a–c); only female mice were used for cognitive assessment and immune profiling following NANA/PBS administration to 5xFAD mice and age-matched WT controls (Fig. 7d–f and Supplementary Fig. 16a–e). When animals from different cohorts/experiments were merged for presentation, the number of cohorts/experiments considered is indicated in the Figure legends. For comorbidity studies (Figs. 1–5, Supplementary Figs. 1–12), the data herein presented originated from five independent cohorts. Animals used for the mass cytometry analysis of the blood immune profile (Supplementary Fig. 6a–c) were selected from the first cohort; animals used for the sNuc-Seq of the hippocampus (Fig. 2a–m, Supplementary Figs. 4 and 5) were selected from all five cohorts; animals used for the sNuc-Seq of the VAT (Fig. 5a–d, Supplementary Figs. 11 and 12) were selected from the first three cohorts; animals used for all other analyses were selected from the last two cohorts. For the study on the effects of NANA administration on the spleen immune profile of young-adult and middle-aged WT mice (Fig. 7a–c and Supplementary Fig. 15a–c), four independent experiments were conducted: two with young-adult mice (6.5-9 mo), and two with middle-aged (11-14 mo) mice. For the study of the effects of NANA administration on novelty discrimination (NOR test; Fig. 7d, e and Supplementary Fig. 16a–c), four independent experiments were conducted using 9-12-mo female mice. Of these four experiments, one could not be included due to complete loss of object discrimination in the PBS-injected 5xFAD controls at the time when the animals were tested (12 mo), which could not allow us to detect further exacerbation by the treatment. No animal was excluded from analyses, except those removed before experimental endpoint according to IACUC guidelines, or because of technical reasons detailed as follows: for microscopic image analysis, poorly stained or overstained sections or slides were not included; for flow and mass cytometry, samples with not enough cells to proceed with the analysis or samples in which the staining did not work were not included; for the analysis of the effects of NANA administration on the spleen immune profile of middle-aged WT mice (Fig. 7c and Supplementary Fig. 15c), one PBS-injected animal was not included due to its statistics across most immune cell populations (1.5x InterQuartile Range method); for the analysis of the effects of NANA administration on novelty discrimination (NOR test; Fig. 7e and Supplementary Fig. 16a–c), one PBS-injected WT mouse was not included in all tested behavioral parameters due to freezing during the test trial of the NOR assay; for the analysis of the effects of NANA administration on the spleen CD4$^+$ T-cell profile of 5xFAD mice (Fig. 7f), one PBS-injected mouse was not included due to its statistics across all immune cell populations (1.5x InterQuartile Range

method). For in vitro studies with mouse lymphocytes, two independent experiments were conducted: one simultaneously testing two concentrations of NANA, 1 and 5 mg/mL (Supplementary Fig. 13c); and one testing 1 mg/mL of NANA. For in vitro studies with human lymphocytes, we performed one pilot study using 10 and 25 mg/mL of NANA. We observed that 25 mg/mL of NANA suppressed both CD4$^+$ and CD8$^+$ T-cell proliferation, whereas 10 mg/mL of NANA had no effect. Since the data from mouse studies in vivo (HFD-fed 5xFAD mice, Fig. 3 and Supplementary Fig. 7; NANA-injected mice, Fig. 7a–c and Supplementary Fig. 15) and in vitro (NANA-treated pan-T-cell cultures, Supplementary Fig. 13) suggested a specific impact on CD4$^+$ T cells, with only a minor effect on CD8$^+$ T cells, in a subsequent single experiment we used an intermediate concentration of 12.5 mg/mL of NANA (Fig. 6 and Supplementary Fig. 14).

### Statistical analyses
The normality of data distribution was evaluated using Anderson-Darling's, D'Agostino-Pearson's ("*omnibus* K2"), Shapiro-Wilk's, and Kolmogorov-Smirnov's tests, and via visual assessment of quantile-quantile plots. Homogeneity of variance was tested using *F*-test, for two groups, and Spearman's test for heteroscedasticity, followed by visual assessment of the homoscedasticity plot, for more than two groups. Data were analyzed using two-tailed Student's *t* test to compare between two groups; Welch's correction was applied for heteroscedastic groups. For lymphocyte cultures, paired Student's *t* test or one-way within-subjects ANOVA followed by Fisher's LSD post hoc test were used. For comorbidity studies, in the presence of two categorical independent variables ("genotype", two levels: "WT" and "5xFAD"; "diet", two levels: "CD" and "HFD"), two-way ANOVA followed by Fisher's LSD post hoc test was used. For weight gain and glucose tolerance test, two-way within-subjects ANOVA followed by Fisher's LSD post hoc test was used. For the analysis of novelty discrimination and locomotor activity/anxiety of NANA-injected 5xFAD mice and PBS-injected 5xFAD and WT controls, one-way ANOVA followed by Fisher's LSD post hoc test was used. For null hypothesis testing, the test statistics with confidence intervals, degrees of freedom, and *P*-values are reported in the Source Data file. For Student's *t* tests and post hoc tests, *P*-values <0.060 are reported in the graphs, rounded to three decimal digits; *P*-value <0.050 was considered significant. Statistical analyses were carried out using GraphPad Prism version 9.0, R, and Microsoft Excel. Graphs were generated with GraphPad Prism version 9.0 and R.

### Reporting summary
Further information on research design is available in the Nature Portfolio Reporting Summary linked to this article.

## Data availability
Source data used for the generation of Figures are provided as a Source Data file. The list of the metabolites identified after plasma metabolite profiling, the metabolite identification criteria, and their relative abundance are included in the Source Data file ("identified metabolites" tab). The raw and processed sequencing data generated in this study are publicly available and have been deposited in the Gene Expression Omnibus database under accession code GSE197082. The accession code is a SuperSeries comprising both the single-nucleus RNA-seq data (SubSeries GSE198835) and the bulk RNA-seq data (SubSeries GSE198144). All other raw data are available from the corresponding authors upon request. Source data are provided with this paper.

## Code availability
The relevant codes used for computational analysis are available at https://github.com/naomihabiblab/HighFatDiet_in_AD.

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

## Acknowledgements

The authors thank Ron Rotkopf from the Bioinformatics unit, Calanit Raanan from the Histology and Pathology unit, Chiara Burgaletto and Davide Raffaele Ceratti from the Weizmann Institute, Michal Bronstein and her team from the Genomics Facility at the Hebrew University, and Julia Waldman and Dan Dubinsky from the Broad Institute for technical support. The work was supported by: the Advanced European Research Council grants 232835 and 741744, the European Seventh Framework Program HEALTH-2011 (279017), the Israel Science Foundation (ISF)—research grant no. 991/16, the ISF-Legacy Heritage Bio-medical Science Partnership research grant no. 1354/15, and the Thompson Foundation and Adelis Foundation (given to M. Schwartz); the Israel Science Foundation (ISF) research grant no. 1709/19, the European Research Council grant 853409, the MOST-IL-China research grant no. 3-15687, and the Myers Foundation (given to N.H.); the National Institutes of Health (NIH) grants DK095045 and DK099465, the Cure Alzheimer's Fund, and the Chan Zuckerberg Foundation (given to A.G.); the Vera and John Schwartz Family Center for Metabolic Biology (given to M.I. and S. Malitsky). N.H. holds the Goren-Khazzam chair in Neuroscience.

## Author contributions

S.S., T.C., A.G., N.H., and M. Schwartz designed, performed, and interpreted the experiments, and wrote the manuscript; S.S., S. Medina, T.C., and S.P.C. performed animal experiments and harvested tissues; S.S. and S. Medina performed behavioral experiments and metabolic assessment; T.C. and S.S. performed flow cytometry and mass cytometry; T.C. performed analysis of flow cytometry data and computational analysis of CyTOF data; T.C. performed in vitro experiments with freshly isolated human PBMCs and mouse splenocytes; T.M.S. performed quality control and tuning of the Helios™ II CyTOF® system; S.P.C. performed immunohistochemistry, imaging, and quantification; S.S. and L.C. performed Aβ1-42 ELISA; D.K. prepared RNA-Seq libraries of hippocampal samples and optimized batches; M.A. assisted in the hippocampal RNA-Seq libraries preparation, optimized nuclei isolation protocol, performed sequencing and initial pre-processing of data; A.R. and O.G. analyzed the hippocampal RNA-Seq data; S. Malitsky and M.I. performed the metabolite profiling; A.B. and T.M. measured NANA in the hippocampus; S.S. analyzed the metabolite profiling data; K.A.V. and M. Slyper optimized the adipose nuclei isolation protocol; K.A.V., M. Slyper, and A.R.C. isolated adipose nuclei; A.R.C., E.K., and A.R. analyzed adipose RNA-Seq data; M.A. performed the bulk RNA-Seq libraries of human lymphocyte culture and initial pre-processing of data; I.S.

analyzed the bulk RNA-Seq data; S.S. performed statistics; S.S., A.R., and T.C. created figures; N.H. guided the single nuclei and bulk RNA-Seq experiments and analysis; A.G. guided the adipose RNA-Seq experiment; M. Schwartz, N.H., and A.G. conceived the study, supervised participants, and secured funding for the study.

## Competing interests

M. Schwartz is a consultant of ImmunoBrain Checkpoint LTD. The remaining authors declare no competing interests.
