## [Peer Review File · Nature Communications]

N-acetylneuraminic acid links immune exhaustion and accelerated memory deficit in diet-induced obese Alzheimer's disease mouse modelREVIEWER COMMENTS

Reviewer #1 (Remarks to the Author):

In the manuscript entitled "Accelerated cognitive decline in obese mouse model of Alzheimer's disease is linked to sialic acid-driven immune deregulation" by Suzzi et al, the authors applied multiple state-of-the-art analytical technologies to assess the effects of high-fat diet on cognitive function, immune regulation (in different body compartments) and disease progression in 5xFAD mice. The study is overall very interesting and has potential impact on the field. However, I have a number of major and minor concerns that should be addressed to improve the manuscript.

Major concerns:

Overall concept of the study:

In summary, the authors have first demonstrated that HFD-induced obesity accelerated cognitive decline in 5xFAD, which was associated with changes in neuronal phenotype, increased neuronal loss, astrogliosis and changes in then cellular landscape of the hippocampus. No results regarding infiltrating immune cells (some were mentioned in the discussion) and specific changes in microglia population were mentioned here. Subsequently, the authors showed the findings on systemic immune deregulation after HFD in isolated splenocyte population. No information about how HFD affects the circulating immune cells in the peripheral blood, the population that are more likely to be in contact with the brain than do the splenocytes. Later, plasma metabolites were quantified, and the lipid-associated macrophages were identified as a source of NANA. This molecule was demonstrated to have influence on mouse splenocytes (in vivo and in vitro) and human blood cells. And no information about brain pathology in NANA-treated mice was shown. Due to this experimental design, several questions remain open and should be shown or (for some) at least discussed in the manuscript. These are for example:

1. How CNS immune cells (e.g. perivascular macrophages and microglia) are affected after HFD, and whether these can be related to changes in neuronal and astroglia population? The authors have briefly mentioned this by citing their own previous papers but the results from current study are missing.
2. Whether the phenotype and function of circulating immune cells in the peripheral blood (similar to splenocytes) are changed after HFD, and whether this is correlated with increased NANA in plasma?
3. Similar to 2. for in vivo and in vitro experiments using NANA administration. It should be also possible to co-culture mouse microglia with NANA and assess the phenotypic or functional changes.
4. Whether NANA treatment in WT or 5xFAD mice can induced similar brain pathology, observed in HFD-treated 5xFAD?

Minor concerns:

1. Based on the results obtained from this study, one would be very curious to know whether NANA can cross the BBB? Or the immune modulation effects observed occur in the periphery and modulated immune cells are the one that drive changes in the CNS? This should be at least discussed.
2. The authors named "splenocytes" as "PBMCs" several time in the manuscript. To my understanding "splenocytes" are not "PBMCs" and possess quite different cellular composition and phenotypes.
3. Line 417:To minimize inter-sample staining variability, sample handling time, and antibody consumption, cells were barcoded with Cell-ID 20-Plex (Fluidigm).... This is not a strategy to minimize inter-sample staining variability.
4. Since the authors used "targeted" approach for the quantification of metabolites, would it be possible that some metabolites, which are related to HFD and have strong effects will not be detected here?
5. Is there any specific reason, why sNuc-seq was used instead of scRNA-Seq?

Reviewer #2 (Remarks to the Author):

This is an interesting manuscript that aims to understand how obesity induced by high fat diet (HFD) contributes to progression of AD and accelerates cognitive decline. To this end, 5XFAD mice were fed a HFD and compared to 5XFAD on control diet (CD) and WT mice on HFD and CD5XFAD on HFD. The manuscript shows that obese 5XFAD mice (with what appears to be 10-15% more weight than control mice) and did worse in the novel object recognition (NOR) test. Obese 5XFAD mice had similar A β and plaque loads but had worse neuronal loss than the other groups. To attempt to understand these findings, the manuscript shows that there is shrinkage of the naive CD4+ T-cells population and expansion of TEMs and Tregs, while TCRb+CD4 – T cells were unaffected in the spleen of obese 5XFAD mice. These mice also had the highest frequency of exhausted TEMs with decreased IFN γ and TNF and increased PD1 and LAG3. To further understand the mechanism of such TEM exhaustion, polar metabolite analysis showed that NANA was increased and that the most likely source was the macrophages in visceral adipose tissues. Short term administration of NANA to middle aged WT mice led to increased CD4+PD1+ T cells and disrupts T cells metabolism. The manuscript concludes that NANA is the diet-associated metabolite associated with both the cognitive decline and the immune deregulation in obese 5x FAD mice, and that chronic antigenic stress and systemic inflammation, found in both obesity and AD, increased immune vulnerability to NANA in obese-AD mice.

Overall, the concepts advanced in the manuscript are relatively novel, and the manuscript is well written, with logical flow. Most of the experiments are well designed and the resulting data support most of the conclusions. However, the manuscript requires additional data to directly prove the most interesting point raised which is that NANA accelerates cognitive decline in obese 5XFAD mice via enhancing TEM exhaustion.

Following are my major concerns:

1-Only one test (NOR) is used to assess cognitive decline in 5XFAD mice in figures 1 and extended data Figure 2. This is hardly the norm. At least one additional test such as Morris water maze, radial arm maze or contextual fear conditioning should be performed to make the case. All of these are well established tests that work with 5XFAD mice.

2-it is not clear why frozen splenocytes were used for analysis of the peripheral immune system, why not using circulating blood immune cells. At least the data from splenocytes should be validated by some analysis of circulating blood lymphocytes.

3-The administration of NANA to middle aged WT mice is interesting. However, the more direct and conclusive experiment is to administer NANA to 5XFAD mice fed control diet and show that the effects of HFD are reproduced by NANA administration including T cell exhaustion as well as cognitive decline and neuronal loss. Without this experiment the manuscript only shows that there is an association between obesity NANA, T cell exhaustion and accelerated neuronal loss and cognitive decline. This is the most relevant experiment that was not included in the manuscript and should be included.

4-It is not clear why peripheral immune exhaustion would cause neuronal loss and cognitive decline. In the absence of any experimental evidence, a detailed discussion is warranted and a cartoon explaining the pathway would be helpful (in addition to the experiments in point # 3 above)

Minor point:

1-from how many mice (from each group) were the 269,578 nuclei that were included in the dataset derived?

Reviewer #3 (Remarks to the Author):

The authors present a manuscript entitled "Accelerated cognitive decline in obese mouse model of Alzheimer's disease is linked to sialic acid-driven immune deregulation" in which they demonstrate using a variety of confirmatory techniques that diet-induced obesity accelerated cognitive loss in 5xFAD mice. The authors also report that the comorbidity effect was functionally linked to systemic immune deregulation and that this was associated with diet-induced elevation of NANA. NANA or N-acetylneuraminic acid is a metabolite identified by the authors using an untargeted metabolomics platform to be inversely correlated with cognitive performance. I found the manuscript to be extremely well written and the inclusion of numerous confirmatory platforms refreshing. I do have some concerns about the metabolomics platform used herein and the number of samples the authors used to come to their conclusions. I bring these comments up as a lot of their manuscript is reliant on the NANA story. I detail them below:

I would like to see a table listing all the "identified" metabolites in the extended data along with their relative concentrations. Further, I would also like to see the authors employ the Metabolites Standards Initiative scoring system for all the recorded metabolites.

I would suggest including the fragmentation pattern for NANA and also include that of the standard to show how they match up in terms of RT and M/Z.

I am very skeptical of the "n" used to identify NANA as being particularly important. The largest group is n=6 and for anyone who conducts metabolomics-based experiments, in particular untargeted metabolomics, having 4-6 biological replicates is very small indeed. I am not sure the correlation coefficients of NANA levels with NOR and Tregs are that strong either with the largest being 0.651.

Figure 3a, I would suggest including another box around "NANA". Maybe in blue, as it is hard to see from the figure which column you are referring to. Is there really that big of a difference across groups? Maybe it is my difficulty in discerning the column.

I would also like to see if the results are recapitulated in another mouse model of AD. Not using all the experimental platforms but maybe a few.

Reviewer #1 (Remarks to the Author):

In the manuscript entitled “Accelerated cognitive decline in obese mouse model of Alzheimer’s disease is linked to sialic acid-driven immune deregulation” by Suzzi et al, the authors applied multiple state-of-the-art analytical technologies to assess the effects of high-fat diet on cognitive function, immune regulation (in different body compartments) and disease progression in 5xFAD mice. The study is overall very interesting and has potential impact on the field.

We thank the Reviewer for the overall positive feedback.

However, I have a number of major and minor concerns that should be addressed to improve the manuscript.

Major concerns:

Overall concept of the study:

In summary, the authors have first demonstrated that HFD-induced obesity accelerated cognitive decline in 5xFAD, which was associated with changes in neuronal phenotype, increased neuronal loss, astrogliosis and changes in then cellular landscape of the hippocampus. No results regarding infiltrating immune cells (some were mentioned in the discussion) and specific changes in microglia population were mentioned here. Subsequently, the authors showed the findings on systemic immune deregulation after HFD in isolated splenocyte population. No information about how HFD affects the circulating immune cells in the peripheral blood, the population that are more likely to be in contact with the brain than do the splenocytes. Later, plasma metabolites were quantified, and the lipid-associated macrophages were identified as a source of NANA. This molecule was demonstrated to have influence on mouse splenocytes (in vivo and in vitro) and human blood cells. And no information about brain pathology in NANA-treated mice was shown. Due to this experimental design, several questions remain open and should be shown or (for some) at least discussed in the manuscript. These are for example:

1. How CNS immune cells (e.g. perivascular macrophages and microglia) are affected after HFD, and whether these can be related to changes in neuronal and astroglia population? The authors have briefly mentioned this by citing their own previous papers but the results from current study are missing.

We now included a more detailed description of the changes in the hippocampal cell landscape, including sub-clustering analysis of the major non-neuronal populations. We show comorbidity-induced transcriptional changes in microglia and other glial cells. Although these are minor changes, these effects are beyond the observed transcriptional switches induced by the genetic model of AD (Results, lines 93-134; Fig. 2a–m; Extended Data Fig. 4a–f, 5a–g). Of note, for perivascular macrophages, we were able to capture this rare cell population in our sequencing data, and found a reduction in their frequencies in the 5xFAD model, but no significant additional change in their proportions with the diet (Fig. 2e, f). On the other hand, we found an increase in T cells specifically in the comorbidity

model (Fig. 2e, f; differentially upregulated genes compared to all other immune cell clusters in Extended Data Table 2).

2. Whether the phenotype and function of circulating immune cells in the peripheral blood (similar to splenocytes) are changed after HFD, and whether this is correlated with increased NANA in plasma?

We now added results of a CyTOF study in which we analyzed the effect of HFD on blood immune profile, and found that the major trend was the increase of CD4⁺ T cells irrespective of genotype (Extended Data Fig. 6a–c in the revised manuscript; relevant text at lines 137-140). This result led us to focus on lymphocytes, and on CD4⁺ T cells, in particular. In subsequent experiments we thus focused on the T-cell profile in the spleen as this organ is a major source of lymphocytes, and deregulation within the splenic CD4⁺ T-cell compartment was previously linked to aging^{1,2} and neurodegeneration^{3,4} (see text, lines 141-143).

3. It should be also possible to co-culture mouse microglia with NANA and assess the phenotypic or functional changes.

In our revised manuscript we added results showing that the microglia were marginally affected by the HFD (Fig. 2e, f, k; relevant text at lines 106-109 and 126-128). Moreover, we show that the levels of NANA in the hippocampus were not different across diet and genotype groups (Extended Data Fig. 10d; relevant text at lines 183-185). Based on these findings, we focused the study on the effect of NANA on T-cells and not on microglia. This is now clarified in the Results (lines 219-221).

3. Whether NANA treatment in WT or 5xFAD mice can induced similar brain pathology, observed in HFD-treated 5xFAD?

In the revised manuscript, we have results showing that repeated NANA administration led to accelerated loss of cognitive performance in the NOR test in 5xFAD mice (Fig. 6k, l; Extended Data Fig. 16a–c; relevant text at lines 250-255). We therefore conclude that exposure to NANA exacerbated the disease phenotype in highly vulnerable mice (5xFAD mice). Please note that identifying the time window of high vulnerability, but when 5xFAD mice were not yet cognitively impaired, was a significant challenge that we overcame in this study. This is now highlighted in the Methods, lines 892-896.

Minor concerns:

4. Based on the results obtained from this study, one would be very curious to know whether NANA can cross the BBB? Or the immune modulation effects observed occur in the periphery

and modulated immune cells are the one that drive changes in the CNS? This should be at least discussed.

NANA can enter the brain from the blood, but this amount is minimal, and the majority of it is cleaved by lyase activity⁵. As noted above (point 3), we measured NANA levels in the hippocampus, and found no difference across diet and genotype groups. Therefore, we hypothesized, based on our results, that the major effects of NANA on disease manifestation in 5xFAD mice is not due to its direct effect on the brain, but rather via its effect on the peripheral immune system. This is now clarified in the text, as suggested by the Reviewer (Discussion, lines 310-318). We also discuss a potential mechanism at lines 279-297.

5. The authors named “splenocytes” as “PBMCs” several time in the manuscript. To my understanding “splenocytes” are not “PBMCs” and possess quite different cellular composition and phenotypes.

We agree with the Reviewer and we now consistently refer to “splenocytes” where relevant throughout the text.

6. Line 417:To minimize inter-sample staining variability, sample handling time, and antibody consumption, cells were barcoded with Cell-ID 20-Plex (Fluidigm).... This is not a strategy to minimize inter-sample staining variability.

Now rephrased (lines 503-505).

7. Since the authors used “targeted” approach for the quantification of metabolites, would it be possible that some metabolites, which are related to HFD and have strong effects will not be detected here?

It is indeed a possibility, since we performed targeted metabolite profiling based on a library of injected standards, which mainly contained *polar* metabolites, and did not investigate *non-polar* metabolites. This point is now noted in the Discussion, lines 300-301.

8. Is there any specific reason, why sNUC-seq was used instead of scRNA-Seq?

In contrast to scRNA-Seq, sNuc-Seq can be used on tissues and cell types that cannot be easily dissociated. In order to get the full landscape of the cells of the hippocampus and visceral adipose tissue, we had to use nuclei, and not cells, because multiple cells are lost upon dissociation, while it remains possible to isolate their nuclei.

Reviewer #2 (Remarks to the Author):

This is an interesting manuscript that aims to understand how obesity induced by high fat diet (HFD) contributes to progression of AD and accelerates cognitive decline. To this end, 5XFAD mice were fed a HFD and compared to 5XFAD on control diet (CD) and WT mice on HFD and CD5XFAD on HFD. The manuscript shows that obese 5XFAD mice (with what appears to be 10-15% more weight than control mice) and did worse in the novel object recognition (NOR) test. Obese 5XFAD mice had similar A β and plaque loads but had worse neuronal loss than the other groups. To attempt to understand these findings, the manuscript shows that there is shrinkage of the naive CD4+ T-cells population and expansion of TEMs and Tregs, while TCRb+CD4 – T cells were unaffected in the spleen of obese 5XFAD mice. These mice also had the highest frequency of exhausted TEMs with decreased IFNg and TNF and increased PD1 and LAG3. To further understand the mechanism of such TEM exhaustion, polar metabolite analysis showed that NANA was increased and that the most likely source was the macrophages in visceral adipose tissues. Short term administration of NANA to middle aged WT mice led to increased CD4+PD1+ T cells and disrupts T cells metabolism. The manuscript concludes that NANA is the diet-associated metabolite associated with both the cognitive decline and the immune deregulation in obese 5xFAD mice, and that chronic antigenic stress and systemic inflammation, found in both obesity and AD, increased immune vulnerability to NANA in obese-AD mice.

Overall, the concepts advanced in the manuscript are relatively novel, and the manuscript is well written, with logical flow. Most of the experiments are well designed and the resulting data support most of the conclusions. However, the manuscript requires additional data to directly prove the most interesting point raised which is that NANA accelerates cognitive decline in obese 5XFAD mice via enhancing TEM exhaustion.

We thank the Reviewer for the positive comments and address the concerns raised below.

Following are my major concerns:

1. Only one test (NOR) is used to assess cognitive decline in 5XFAD mice in figures 1 and extended data Figure 2. This is hardly the norm. At least one additional test such as Morris water maze, radial arm maze or contextual fear conditioning should be performed to make the case. All of these are well established tests that work with 5XFAD mice.

In our initial studies we carried out also Radial Arm Water Maze. However, since the goal was to detect a time window in which the HFD accelerated cognitive loss in 5xFAD mice relative to the CD, we had to carry out repeated behavioral tests. We found that the NOR was the least stressful, and therefore continued with this assay throughout the entire work. In addition, we also assessed locomotor activity/anxiety by measuring the total distance moved and the time spent in the middle of the arena during the habituation trial of the NOR test. We now include these results in the revised manuscript (Extended Data Fig. 2d–g). We also report the total exploration time during the test trial (Extended Data Fig. 2b, c). The relevant text is at lines 64-79. The same parameters were tested in the additional data that

we now include related to the effect of treatment with NANA on cognitive performance of 5xFAD mice (Extended Data Fig. 16a–c in the revised manuscript), as discussed below at point 3.

2. It is not clear why frozen splenocytes were used for analysis of the peripheral immune system, why not using circulating blood immune cells. At least the data from splenocytes should be validated by some analysis of circulating blood lymphocytes.

We now clarify the use of *fresh splenocytes* for general flow cytometry analysis (lines 143-144) and *frozen splenocytes* for mass cytometry (CyTOF) analysis (lines 152-154). We also now added CyTOF results of the blood immune profile that shows that the CD4⁺ T cells were impacted by the HFD, regardless of genotype (Extended Data Fig. 6a–c in the revised manuscript; relevant text at lines 137-140). Therefore, we focused in subsequent experiments on lymphocytes and CD4⁺ T cells, and thus analyzed the spleen as this organ is a richer source of lymphocytes, and splenic CD4⁺ T-cell rearrangements were previously shown to be linked to aging^{1,2} and neurodegeneration^{3,4}. With respect to validation of splenocyte data with blood lymphocyte data, we wish to emphasize that the *ex vivo* experiments, where we cultured mouse splenic T cells with or without NANA (Extended Data Fig. 13a–g in the revised manuscript), were validated by the *ex vivo* experiments with human *blood* lymphocytes (Fig. 6a–c and Extended Data Fig. 14a–e in the revised manuscript). This is now clarified in the text (Results, lines 221-225; Discussion, lines 314-316).

3. The administration of NANA to middle aged WT mice is interesting. However, the more direct and conclusive experiment is to administer NANA to 5XFAD mice fed control diet and show that the effects of HFD are reproduced by NANA administration including T cell exhaustion as well as cognitive decline and neuronal loss. Without this experiment the manuscript only shows that there is an association between obesity NANA, T cell exhaustion and accelerated neuronal loss and cognitive decline. This is the most relevant experiment that was not included in the manuscript and should be included.

As we note in our reply to Reviewer #1, point 3, we now have results showing that 5xFAD mice treated with NANA, at 3-4 weeks after repeated NANA injections, displayed accelerated cognitive loss (Fig. 6k, l). In addition, in the same mice we show a CD4⁺ T-cell profile in the spleen and blood similar to the CD4⁺ T-cell profile observed in the spleen of middle-aged WT mice treated with NANA and analyzed 1 day after the last injection, which we reported in our original submission (Fig. 6h, j). In the revised manuscript, we describe these results at lines 250-267; the relevant Figures are Fig. 6h–l, Extended Data Fig. 15a–c, and Extended Data Fig. 16a–f. Finally, we wish to emphasize that both our *ex vivo* and *in vivo* experiments with NANA suggest that NANA has the potential to drive T-cell exhaustion, with CD4⁺ T cells being more susceptible (Fig. 6a–m; Extended Data Fig. 13a–g, Extended Data Fig. 14a–e; Extended Data Fig. 15a–c).

4. It is not clear why peripheral immune exhaustion would cause neuronal loss and cognitive decline. In the absence of any experimental evidence, a detailed discussion is warranted and a cartoon explaining the pathway would be helpful (in addition to the experiments in point # 3 above).

There are numerous studies by our team and others, both in mouse models of aging and AD, and recently in humans, showing a correlation between progression of cognitive loss and reduction of IFN- γ production as well as elevation of Tregs in the periphery (for example, references^{3,4,6-13}). This issue is now thoroughly discussed, and all the relevant citations are included in the revised manuscript, lines 279-297.

Minor point:

1. from how many mice (from each group) were the 269,578 nuclei that were included in the dataset derived?

The reported nuclei were derived from 28 animals, across all mouse genotype and diet groups (WT CD, WT HFD, 5xFAD CD, and 5xFAD HFD). Of note, in the revised manuscript we have 269,503 nuclei as few additional nuclei were removed due to their lower quality. The relevant text is in the Methods, lines 721-735.

Reviewer #3 (Remarks to the Author):

The authors present a manuscript entitled “Accelerated cognitive decline in obese mouse model of Alzheimer’s disease is linked to sialic acid-driven immune deregulation” in which they demonstrate using a variety of confirmatory techniques that diet-induced obesity accelerated cognitive loss in 5xFAD mice. The authors also report that the comorbidity effect was functionally linked to systemic immune deregulation and that this was associated with diet-induced elevation of NANA. NANA or N-acetylneuraminic acid is a metabolite identified by the authors using an untargeted metabolomics platform to be inversely correlated with cognitive performance. I found the manuscript to be extremely well written and the inclusion of numerous confirmatory platforms refreshing.

We thank the Reviewer for the appreciative comments.

I do have some concerns about the metabolomics platform used herein and the number of samples the authors used to come to their conclusions. I bring these comments up as a lot of their manuscript is reliant on the NANA story. I detail them below:

1. I would like to see a table listing all the “identified” metabolites in the extended data along with their relative concentrations. Further, I would also like to see the authors employ the Metabolites Standards Initiative scoring system for all the recorded metabolites.

We now provide as requested (Extended Data Table 3). Metabolites' abundance is reported as normalized peak area (Methods, lines 571-573).

2. I would suggest including the fragmentation pattern for NANA and also include that of the standard to show how they match up in terms of RT and M/Z.

We now provide representative mass chromatograms and MS/MS spectra of NANA sample and standard (Extended Data Fig. 10a, b). Furthermore, in the revised manuscript we include targeted measurements of NANA levels in the hippocampus (Extended Data Fig. 10d), and provide representative LC-MS/MS chromatograms of NANA standard and ¹³C₃-NANA internal standard (Extended Data Fig. 10e).

2. I am very skeptical of the “n” used to identify NANA as being particularly important. The largest group is n=6 and for anyone who conducts metabolomics-based experiments, in particular untargeted metabolomics, having 4-6 biological replicates is very small indeed. I am not sure the correlation coefficients of NANA levels with NOR and Tregs are that strong either with the largest being 0.651.

As already reported in the original submission, we used a fluorometric assay to measure free NANA in a larger number of plasma samples, including samples from the same animals used for untargeted metabolomics (Fig. 4e in the revised manuscript). Using these measurements, we now show correlation of NANA levels with NOR discrimination index (Extended Data Fig. 10f) and splenic Tregs frequency (Extended Data Fig. 10g), in line with our findings from untargeted metabolomics (Fig. 4c and d, respectively). The relevant text is at lines 1845-192.

3. Figure 3a, I would suggest including another box around “NANA”. Maybe in blue, as it is hard to see from the figure which column you are referring to. Is there really that big of a difference across groups? Maybe it is my difficulty in discerning the column.

Based on the Reviewer's suggestion, we modified the relevant Figure (Fig. 4a in the revised manuscript). In addition, we now show the amount of NANA per sample from untargeted metabolomics in Extended Data Fig. 10c.

4. I would also like to see if the results are recapitulated in another mouse model of AD. Not using all the experimental platforms but maybe a few.

There are several publications supporting the relationships between obesity and AD¹⁴⁻¹⁹, between obesity and immune dysfunction²⁰⁻²², and between immune dysfunction and

cognitive decline/AD^{3,4,9,23-28}. Based on this evidence, our findings argue in favor of a more general phenomenon that is independent of the primary cause of the disease.

References

1. Han, G.-M., Zhao, B., Jeyaseelan, S. & Feng, J.-M. Age-associated parallel increase of Foxp3(+)CD4(+) regulatory and CD44(+)CD4(+) memory T cells in SJL/J mice. *Cell. Immunol.* **258**, 188–196 (2009).
2. Elyahu, Y. *et al.* Aging promotes reorganization of the CD4 T cell landscape toward extreme regulatory and effector phenotypes. *Sci. Adv.* **5**, eaaw8330 (2019).
3. Baruch, K. *et al.* Breaking immune tolerance by targeting Foxp3(+) regulatory T cells mitigates Alzheimer's disease pathology. *Nat. Commun.* **6**, 7967 (2015).
4. Wang, F. *et al.* Splenocytes derived from young WT mice prevent AD progression in APP^{swe}/PSEN1^{dE9} transgenic mice. *Oncotarget* **6**, 20851–20862 (2015).
5. Nöhle, U. & Schauer, R. Uptake, metabolism and excretion of orally and intravenously administered, 14C- and 3H-labeled N-acetylneuraminic acid mixture in the mouse and rat. *Hoppe. Seylers. Z. Physiol. Chem.* **362**, 1495–1506 (1981).
6. Filiano, A. J. *et al.* Unexpected role of interferon- γ in regulating neuronal connectivity and social behaviour. *Nature* **535**, 425–429 (2016).
7. Baruch, K. *et al.* Aging. Aging-induced type I interferon response at the choroid plexus negatively affects brain function. *Science* **346**, 89–93 (2014).
8. He, Z. *et al.* Intraperitoneal injection of IFN- γ restores microglial autophagy, promotes amyloid- β clearance and improves cognition in APP/PS1 mice. *Cell Death Dis.* **11**, 440 (2020).
9. Yang, H.-S. *et al.* Plasma IL-12/IFN- γ axis predicts cognitive trajectories in cognitively unimpaired older adults. *Alzheimers. Dement.* **18**, 645–653 (2022).
10. Mesquita, S. D. *et al.* The choroid plexus transcriptome reveals changes in type I and II interferon responses in a mouse model of Alzheimer's disease. *Brain. Behav. Immun.* **49**, 280–292 (2015).
11. Baruch, K. *et al.* PD-1 immune checkpoint blockade reduces pathology and improves memory in mouse models of Alzheimer's disease. *Nat. Med.* **22**, 135–137 (2016).
12. Rosenzweig, N. *et al.* PD-1/PD-L1 checkpoint blockade harnesses monocyte-derived macrophages to combat cognitive impairment in a tauopathy mouse model. *Nat. Commun.* **10**, 465 (2019).
13. Liu, H., Zhao, J., Lin, Y., Su, M. & Lai, L. Administration of anti-ERMAP antibody ameliorates Alzheimer's disease in mice. *J. Neuroinflammation* **18**, 268 (2021).
14. Alford, S., Patel, D., Perakakis, N. & Mantzoros, C. S. Obesity as a risk factor for Alzheimer's disease: weighing the evidence. *Obes. Rev. an Off. J. Int. Assoc. Study Obes.* **19**, 269–280 (2018).
15. Dake, M. D. *et al.* Obesity and Brain Vulnerability in Normal and Abnormal Aging: A Multimodal MRI Study. *J. Alzheimer's Dis. Reports* **5**, 65–77 (2021).
16. O'Brien, P. D., Hinder, L. M., Callaghan, B. C. & Feldman, E. L. Neurological consequences of obesity. *Lancet. Neurol.* **16**, 465–477 (2017).

17. Chuang, Y.-F. *et al.* Midlife adiposity predicts earlier onset of Alzheimer's dementia, neuropathology and presymptomatic cerebral amyloid accumulation. *Mol. Psychiatry* **21**, 910–915 (2016).
18. Singh-Manoux, A. *et al.* Obesity trajectories and risk of dementia: 28 years of follow-up in the Whitehall II Study. *Alzheimers. Dement.* **14**, 178–186 (2018).
19. Dahl, A. K. & Hassing, L. B. Obesity and Cognitive Aging. *Epidemiol. Rev.* **35**, 22–32 (2013).
20. Aguilar, E. G. & Murphy, W. J. Obesity induced T cell dysfunction and implications for cancer immunotherapy. *Curr. Opin. Immunol.* **51**, 181–186 (2018).
21. Wang, Z. *et al.* Paradoxical effects of obesity on T cell function during tumor progression and PD-1 checkpoint blockade. *Nat. Med.* **25**, 141–151 (2019).
22. Franceschi, C., Garagnani, P., Parini, P., Giuliani, C. & Santoro, A. Inflammaging: a new immune-metabolic viewpoint for age-related diseases. *Nat. Rev. Endocrinol.* **14**, 576–590 (2018).
23. Marsh, S. E. *et al.* The adaptive immune system restrains Alzheimer's disease pathogenesis by modulating microglial function. *Proc. Natl. Acad. Sci. U. S. A.* **113**, E1316–E1325 (2016).
24. El Khoury, J. *et al.* Ccr2 deficiency impairs microglial accumulation and accelerates progression of Alzheimer-like disease. *Nat. Med.* **13**, 432–438 (2007).
25. Minhas, P. S. *et al.* Restoring metabolism of myeloid cells reverses cognitive decline in ageing. *Nature* **590**, 122–128 (2021).
26. Zhao, Y., Zhan, J.-K. & Liu, Y. A Perspective on Roles Played by Immunosenescence in the Pathobiology of Alzheimer's Disease. *Aging Dis.* **11**, 1594–1607 (2020).
27. Lutshumba, J., Nikolajczyk, B. S. & Bachstetter, A. D. Dysregulation of Systemic Immunity in Aging and Dementia. *Front. Cell. Neurosci.* **15**, 652111 (2021).
28. Da Mesquita, S. *et al.* Functional aspects of meningeal lymphatics in ageing and Alzheimer's disease. *Nature* **560**, 185–191 (2018).

REVIEWER COMMENTS

Reviewer #1 (Remarks to the Author):

The authors have revised the manuscript and updated the figures to include additional information. The revised version of the manuscript addresses all of my comments. The work presented by the authors is valuable to the scientific community and will support other studies, especially those in AD research.

Reviewer #3 (Remarks to the Author):

The authors have addressed all of my comments and concerns.

Reviewer #4 (Remarks to the Author):

Reviewer 2

Major points

Response 1:

We appreciate the comment of the authors and the extra included effects of treatment with NANA on the cognitive performance of 5xFAD mice. However, the first point raised by Reviewer 2 has not been sufficiently addressed. The authors merely comment that it is not necessary to include at least one additional behavioral test since (1) they had to carry it out throughout the entire workflow (causing stress for the animals), and the chosen test was the least stressful (2) they now show extra data (locomotor activity/anxiety by measuring total distance moved and time spent in the middle of the arena (of the same test). However, when the 'time window' for the cognitive decline was determined using the NOR test, authors could have decided to, only at that point, also include other established tests for cognitive decline. We do understand that it is not easy to repeat this for a longitudinal mouse experiment of this length. But, if the authors cannot show cognitive decline using different cognitive tests, and the effects on cognition are not that pronounced, the authors should at least rephrase the title, which has a major focus on cognitive decline.

Response 2:

Well-resolved.

Response 3:

Well-resolved.

Response 4:

We appreciate the more thorough discussion about correlations between T cells and neuronal loss / cognitive decline. However, we still feel that the authors do not explain a direct molecular mechanism. In addition, the authors hypothesize that increased T cell senescence could lead to diminished levels of IFN- γ which would induce cognitive decline. If so, why did the authors not measure and compare peripheral IFN- γ levels? In addition, the authors only cite and discuss papers that provide evidence for the hypothesis that Th1 cytokines promote beneficial neuronal function. However, there is also much evidence showing the opposite. In addition, Latta-Mahieu et al (2017) showed that blocking immune senescence using immunotherapy against PD-1 is not sufficient to reduce amyloid pathology in APP/PS1 mice. We would prefer a more honest representation of the literature in this paragraph (279-297). In addition, it would be key for their hypothesis if the authors could show that NANA reduces IFN- γ levels in the periphery and at the choroid plexus in the 5xFAD mice and that this correlates to

accelerated cognitive decline. If this is not feasible, the authors could investigate if stimulating splenic CD4+ T cells in vitro with NANA reduces their IFN- γ release upon stimulation.

Minor points

Response 1:

Resolved.

Point-By-Point Response

Reviewer #1 (Remarks to the Author):

The authors have revised the manuscript and updated the figures to include additional information. The revised version of the manuscript addresses all of my comments. The work presented by the authors is valuable to the scientific community and will support other studies, especially those in AD research.

We thank the Reviewer for the kind words.

Reviewer #3 (Remarks to the Author):

The authors have addressed all of my comments and concerns.

We thank the Reviewer for the positive feedback.

Reviewer #4 (Remarks to the Author):

Reviewer 2

Major points

Response 1:

We appreciate the comment of the authors and the extra included effects of treatment with NANA on the cognitive performance of 5xFAD mice. However, the first point raised by Reviewer 2 has not been sufficiently addressed. The authors merely comment that it is not necessary to include at least one additional behavioral test since (1) they had to carry it out throughout the entire workflow (causing stress for the animals), and the chosen test was the least stressful (2) they now show extra data (locomotor activity/anxiety by measuring total distance moved and time spent in the middle of the arena (of the same test). However, when the ‘time window’ for the cognitive decline was determined using the NOR test, authors could have decided to, only at that point, also include other established tests for cognitive decline. We do understand that it is not easy to repeat this for a longitudinal mouse experiment of this length. But, if the authors cannot show cognitive decline using different cognitive tests, and the effects on cognition are not that pronounced, the authors should at least rephrase the title, which has a major focus on cognitive decline.

We accept this comment and in the revised manuscript we rephrased the title (now “Accelerated disease manifestations in obese mouse model of amyloidosis is linked to sialic acid-driven immune deregulation”) and, throughout the main text, we toned down statements regarding the effect we found on cognition, and relate to these effects as “disease manifestations”.

Response 2:

Well-resolved.

Thanks.

Response 3:

Well-resolved.

Thanks.

Response 4:

We appreciate the more thorough discussion about correlations between T cells and neuronal loss/cognitive decline. However, we still feel that the authors do not explain a direct molecular mechanism. In addition, the authors hypothesize that increased T cell senescence could lead to diminished levels of IFN- γ which would induce cognitive decline. If so, why did the authors not measure and compare peripheral IFN- γ levels? In addition, the authors only cite and discuss papers that provide evidence for the hypothesis that Th1 cytokines promote beneficial neuronal function. However, there is also much evidence showing the opposite. In addition, Latta-Mahieu et al (2017) showed that blocking immune senescence using immunotherapy against PD-1 is not sufficient to reduce amyloid pathology in APP/PS1 mice. We would prefer a more honest representation of the literature in this paragraph (279-297). In addition, it would be key for their hypothesis if the authors could show that NANA reduces IFN- γ levels in the periphery and at the choroid plexus in the 5xFAD mice and that this correlates to accelerated cognitive decline. If this is not feasible, the authors could investigate if stimulating splenic CD4⁺ T cells in vitro with NANA reduces their IFN- γ release upon stimulation.

Perhaps, in our original revised version, our response to Referee #2 was misleading. By no means we meant that the sole effect of the HFD-accelerated disease manifestations, via NANA elevation in the blood, could be due to diminished levels of IFN- γ in the periphery. In fact, we observed HFD-induced changes in multiple aspects of the immune system that together suggested an immune aging-like effect, of which IFN- γ reduction may be one possible, but certainly not the exclusive component. Based on this Reviewer's comments, we now revised the Discussion, focusing our proposed mechanism on the quantitative and qualitative changes within the CD4⁺ T-cell compartment, as we found in HFD-fed 5xFAD mice and experiments with NANA. We now cite work showing that CD4⁺ T cells are specifically implicated in several aspects of brain function¹⁻⁴, and that altered frequencies of CD4⁺ T-cell subsets occur in aging⁵⁻⁸ and in human AD patients⁹⁻¹¹ similarly to what we found in 5xFAD mice under HFD or mice treated with NANA. We further discuss the apparently controversial role of CD4⁺ regulatory T cells (Tregs) in human AD patients^{10,12-18} and in mouse models of AD¹⁹⁻²⁵, arguing that, although Tregs are needed within the brain to resolve damage together with bone marrow-derived myeloid cells¹⁹⁻²⁵, their role in the periphery may be protective or detrimental depending on the disease stage^{23,26,27}. In summary,

we now more clearly link the accelerated disease manifestations in HFD-fed 5xFAD mice to the accelerated aging of the CD4⁺ T cell compartment, at least in part driven by NANA, given that aging impairs CD4⁺ T cell functionality and, thus, the overall ability of the immune system to protect the brain^{8,28}. In line, we now also include the recent studies showing that immune aging is sufficient to drive aging of non-lymphoid solid organs, including the brain²⁹, and is linked to cognitive decline³⁰, and that targeting the exhaustion-related PD-1/PD-L1 pathway elicits general anti-aging effects³¹. Regarding the Reviewer's comment that we did not cite the paper by Latta-Mahieu et al 2018³², we wish to emphasize that we cited in the original manuscript studies that showed that "rejuvenation" of the immune system, achieved by systemic blockade of inhibitory immune checkpoints (PD-1/PD-L1³³⁻³⁶ and, more recently, ERMAP³⁷), led to improved cognition in several mouse models of neurodegeneration. The work referred by this Reviewer did not measure cognitive performance but only showed no differences in amyloid plaques burden, which is less relevant to our study as we did not observe differences in either amyloid β plaques or soluble amyloid β in HFD-fed 5xFAD mice. Nevertheless, we now cite the work by Latta-Mahieu et al 2018³², as the Reviewer suggested.

Minor points

Response 1:

Resolved.

Thanks.

References

1. Pasciuto, E. *et al.* Microglia Require CD4 T Cells to Complete the Fetal-to-Adult Transition. *Cell* **182**, 625-640.e24 (2020).
2. Wolf, S. A. *et al.* CD4-positive T lymphocytes provide a neuroimmunological link in the control of adult hippocampal neurogenesis. *J. Immunol.* **182**, 3979–3984 (2009).
3. Radjavi, A., Smirnov, I. & Kipnis, J. Brain antigen-reactive CD4+ T cells are sufficient to support learning behavior in mice with limited T cell repertoire. *Brain. Behav. Immun.* **35**, 58–63 (2014).
4. Evans, F. L., Dittmer, M., de la Fuente, A. G. & Fitzgerald, D. C. Protective and Regenerative Roles of T Cells in Central Nervous System Disorders. *Front. Immunol.* **10**, 2171 (2019).
5. Han, G.-M., Zhao, B., Jeyaseelan, S. & Feng, J.-M. Age-associated parallel increase of Foxp3(+)CD4(+) regulatory and CD44(+)CD4(+) memory T cells in SJL/J mice. *Cell. Immunol.* **258**, 188–196 (2009).
6. Elyahu, Y. *et al.* Aging promotes reorganization of the CD4 T cell landscape toward extreme regulatory and effector phenotypes. *Sci. Adv.* **5**, eaaw8330 (2019).
7. Harpaz, I., Bhattacharya, U., Elyahu, Y., Strominger, I. & Monsonogo, A. Old Mice Accumulate Activated Effector CD4 T Cells Refractory to Regulatory T Cell-Induced Immunosuppression. *Front. Immunol.* **8**, 283 (2017).
8. Lefebvre, J. S. & Haynes, L. Aging of the CD4 T Cell Compartment. *Open Longev. Sci.* **6**, 83–91 (2012).
9. Larbi, A. *et al.* Dramatic shifts in circulating CD4 but not CD8 T cell subsets in mild Alzheimer’s disease. *J. Alzheimers. Dis.* **17**, 91–103 (2009).
10. Pellicanò, M. *et al.* Immune profiling of Alzheimer patients. *J. Neuroimmunol.* **242**, 52–59 (2012).
11. Wu, C.-T. *et al.* A change of PD-1/PD-L1 expression on peripheral T cell subsets correlates with the different stages of Alzheimer’s Disease. *Cell Biosci.* **12**, 162 (2022).
12. Faridar, A. *et al.* Restoring regulatory T-cell dysfunction in Alzheimer’s disease through ex vivo expansion. *Brain Commun.* **2**, (2020).
13. Oberstein, T. J. *et al.* Imbalance of Circulating T(h)17 and Regulatory T Cells in Alzheimer’s Disease: A Case Control Study. *Front. Immunol.* **9**, 1213 (2018).
14. Ciccocioppo, F. *et al.* The Characterization of Regulatory T-Cell Profiles in Alzheimer’s Disease and Multiple Sclerosis. *Sci. Rep.* **9**, 8788 (2019).
15. Saresella, M. *et al.* PD1 negative and PD1 positive CD4+ T regulatory cells in mild cognitive impairment and Alzheimer’s disease. *J. Alzheimers. Dis.* **21**, 927–938 (2010).
16. Fu, J. *et al.* Mild Cognitive Impairment Patients Have Higher Regulatory T-Cell Proportions Compared With Alzheimer’s Disease-Related Dementia Patients. *Front. Aging Neurosci.* **12**, 624304 (2020).

17. Le Page, A. *et al.* Differential Phenotypes of Myeloid-Derived Suppressor and T Regulatory Cells and Cytokine Levels in Amnesic Mild Cognitive Impairment Subjects Compared to Mild Alzheimer Diseased Patients. *Front. Immunol.* **8**, 783 (2017).
18. Rosenkranz, D. *et al.* Higher frequency of regulatory T cells in the elderly and increased suppressive activity in neurodegeneration. *J. Neuroimmunol.* **188**, 117–127 (2007).
19. Butovsky, O., Kunis, G., Koronyo-Hamaoui, M. & Schwartz, M. Selective ablation of bone marrow-derived dendritic cells increases amyloid plaques in a mouse Alzheimer's disease model. *Eur. J. Neurosci.* **26**, 413–416 (2007).
20. Koronyo, Y. *et al.* Therapeutic effects of glatiramer acetate and grafted CD115⁺ monocytes in a mouse model of Alzheimer's disease. *Brain* **138**, 2399–2422 (2015).
21. Shechter, R. *et al.* Recruitment of beneficial M2 macrophages to injured spinal cord is orchestrated by remote brain choroid plexus. *Immunity* **38**, 555–569 (2013).
22. Raposo, C. *et al.* CNS Repair Requires Both Effector and Regulatory T Cells with Distinct Temporal and Spatial Profiles. *J. Neurosci.* **34**, 10141 LP – 10155 (2014).
23. Baruch, K. *et al.* Breaking immune tolerance by targeting Foxp3(+) regulatory T cells mitigates Alzheimer's disease pathology. *Nat. Commun.* **6**, 7967 (2015).
24. Ben-Yehuda, H. *et al.* Key role of the CCR2-CCL2 axis in disease modification in a mouse model of tauopathy. *Mol. Neurodegener.* **16**, 39 (2021).
25. Ito, M. *et al.* Brain regulatory T cells suppress astrogliosis and potentiate neurological recovery. *Nature* **565**, 246–250 (2019).
26. Dansokho, C. *et al.* Regulatory T cells delay disease progression in Alzheimer-like pathology. *Brain* **139**, 1237–1251 (2016).
27. Baek, H. *et al.* Neuroprotective effects of CD4⁺CD25⁺Foxp3⁺ regulatory T cells in a 3xTg-AD Alzheimer's disease model. *Oncotarget* **7**, 69347–69357 (2016).
28. Kipnis, J., Gadani, S. & Derecki, N. C. Pro-cognitive properties of T cells. *Nature reviews. Immunology* vol. 12 663–669 (2012).
29. Yousefzadeh, M. J. *et al.* An aged immune system drives senescence and ageing of solid organs. *Nature* **594**, 100–105 (2021).
30. Minhas, P. S. *et al.* Restoring metabolism of myeloid cells reverses cognitive decline in ageing. *Nature* **590**, 122–128 (2021).
31. Wang, T.-W. *et al.* Blocking PD-L1–PD-1 improves senescence surveillance and ageing phenotypes. *Nature* **611**, 358–364 (2022).
32. Latta-Mahieu, M. *et al.* Systemic immune-checkpoint blockade with anti-PD1 antibodies does not alter cerebral amyloid- β burden in several amyloid transgenic mouse models. *Glia* **66**, 492–504 (2018).
33. Baruch, K. *et al.* PD-1 immune checkpoint blockade reduces pathology and improves memory in mouse models of Alzheimer's disease. *Nat. Med.* **22**, 135–137 (2016).
34. Rosenzweig, N. *et al.* PD-1/PD-L1 checkpoint blockade harnesses monocyte-derived macrophages to combat cognitive impairment in a tauopathy mouse model. *Nat. Commun.*

- 10**, 465 (2019).
35. Xing, Z. *et al.* Influenza vaccine combined with moderate-dose PD1 blockade reduces amyloid- β accumulation and improves cognition in APP/PS1 mice. *Brain. Behav. Immun.* **91**, 128–141 (2021).
 36. Zou, Y. *et al.* Programmed Cell Death Protein 1 Blockade Reduces Glycogen Synthase Kinase 3 β Activity and Tau Hyperphosphorylation in Alzheimer's Disease Mouse Models. *Front. Cell Dev. Biol.* **9**, (2021).
 37. Liu, H., Zhao, J., Lin, Y., Su, M. & Lai, L. Administration of anti-ERMAP antibody ameliorates Alzheimer's disease in mice. *J. Neuroinflammation* **18**, 268 (2021).

REVIEWERS' COMMENTS

Reviewer #4 (Remarks to the Author):

The authors have sufficiently address our comments.